# Neural crest-specific deletion of *Rbfox2* in mice leads to craniofacial abnormalities including cleft palate

Dasan Mary Cibi[1], Masum M Mia[1], Shamini Guna Shekeran[1], Lim Sze Yun[2], Reddemma Sandireddy[1], Priyanka Gupta[1], Monalisa Hota[1], Lei Sun[1], Sujoy Ghosh[1], Manvendra K Singh[1,2]*

[1]Program in Cardiovascular and Metabolic Disorders, Duke-NUS Medical School, Singapore, Singapore; [2]National Heart Research Institute, National Heart Center, Singapore, Singapore

**Abstract** Alternative splicing (AS) creates proteomic diversity from a limited size genome by generating numerous transcripts from a single protein-coding gene. Tissue-specific regulators of AS are essential components of the gene regulatory network, required for normal cellular function, tissue patterning, and embryonic development. However, their cell-autonomous function in neural crest development has not been explored. Here, we demonstrate that splicing factor Rbfox2 is expressed in the neural crest cells (NCCs), and deletion of *Rbfox2* in NCCs leads to cleft palate and defects in craniofacial bone development. RNA-Seq analysis revealed that Rbfox2 regulates splicing and expression of numerous genes essential for neural crest/craniofacial development. We demonstrate that Rbfox2-TGF-β-Tak1 signaling axis is deregulated by *Rbfox2* deletion. Furthermore, restoration of TGF-β signaling by Tak1 overexpression can rescue the proliferation defect seen in *Rbfox2* mutants. We also identified a positive *feedback loop* in which TGF-β signaling promotes expression of *Rbfox2* in NCCs.
DOI: https://doi.org/10.7554/eLife.45418.001

*For correspondence:
manvendra.singh@duke-nus.edu.sg

Competing interests: The authors declare that no competing interests exist.

## Introduction

Approximately 3–4% of infants are born with congenital diseases. Collectively, craniofacial and cardiovascular abnormalities are the most common defects, contributing to more than one-third of the congenital diseases. Proper formation of these structures involves intricate processes such as proliferation, migration, and differentiation of NCCs, as well as their interaction with neighboring cells (*Martik and Bronner, 2017*; *Plein et al., 2015*). NCCs are a transient population of migratory progenitor cells that reside in the dorsal side of the neural tube. The NCCs separate from their neighboring cells through a delamination process and migrate via different pathways throughout the body to diverse locations, differentiating into multiple cell types at their respective destinations (*Martik and Bronner, 2017*). NCCs can be subdivided into five axial populations: cranial, cardiac, vagal, trunk and sacral NC cells. Lineage-tracing experiments in mouse embryos have demonstrated that NCCs differentiate into a diverse array of cell types including the craniofacial skeletal elements, autonomous nervous system of the heart and GI-tract, as well as smooth muscle cells of the cardiac OFT (*Chai et al., 2000*; *Jiang et al., 2002*; *Jiang et al., 2000*; *Lee et al., 2004*; *Martik and Bronner, 2017*; *Plein et al., 2015*; *Waldo et al., 1999*; *Yoshida et al., 2008*).

Cranial NCCs migrate from anterior portions of the folded neural tube and contribute to the formation of the skull, cartilage, and connective tissue, where they populate the first and second pharyngeal arches that give rise to cranial ganglia, the maxilla, the mandible, palates and other structures of the developing head (*Chai et al., 2000*; *Jiang et al., 2002*; *Martik and Bronner,*

**eLife digest** Abnormalities affecting the head and face – such as a cleft lip or palate – are among the most common of all birth defects. These tissues normally develop from cells in the embryo known as the neural crest cells, and specifically a subset of these cells called the cranial neural crest cells. Most cases of cleft lip or palate are linked back to genes that affect the biology of this group of cells.

The list of genes implicated in the impaired development of cranial neural crest cells code for proteins with a wide range of different activities. Some encode transcription factors – proteins that switch genes on or off. Others code for chromatin remodeling factors, which control how the DNA is packed inside cells. However, the role of another group of proteins – the splicing factors – remains unclear and warrants further investigation.

When a gene is switched on its genetic code is first copied into a short-lived molecule called a transcript. These transcripts are then edited to form templates to build proteins. Splicing is one way that a transcript can be edited, which involves different pieces of the transcript being cut out and the remaining pieces being pasted together to form alternative versions of the final template. Splicing factors control this process.

Cibi et al. now show that neural crest cells from mice make a splicing factor called Rbfox2 and that deleting this gene for this protein from only these cells leads to mice with a cleft palate and defects in the bones of their head and face.

Further analysis helped to identify the transcripts that are spliced by Rbfox2, and the effects that these splicing events have on gene activity in mouse tissues that develop from cranial neural crest cells. Cibi et al. went on to find a signaling pathway that was impaired in the mutant cells that lacked Rbfox2. Forcing the mutant cells to over-produce one of the proteins involved in this signaling pathway (a protein named Tak1) was enough to compensate for the some of the defects caused by a lack of Rbfox2, suggesting it acts downstream of the splicing regulator.

Lastly, Cibi et al. showed that another protein in this signaling pathway, called TGF-β, acted to increase how much Rbfox2 was made by neural crest cells. Together these findings may be relevant in human disease studies, given that altered TGF-β signaling is a common feature in many birth defects seen in humans.

DOI: https://doi.org/10.7554/eLife.45418.002

2017). In mammals, both the primary and secondary palates are morphologically visible as early as E11.5 as demonstrated by an outgrowth from the oral side of the medial nasal and maxillary processes, respectively. The primary palate is formed by fusion of the maxillary prominence with the frontonasal prominence (*Bush and Jiang, 2012*; *Dixon et al., 2011*). Secondary palates are formed from the outgrowths of neural crest-derived mesenchyme that lie on either side of the developing tongue (*Bush and Jiang, 2012*). Initially, palatal shelves grow vertically flanking the developing tongue, but they elevate between E13.5 and E14.5 to a horizontal position above the tongue, grow toward the midline and fuse with each other to form an intact palate (*Bush and Jiang, 2012*). Functional defects in NCCs result in craniofacial malformations including cleft lip and/or cleft palate. Many transcription factors, chromatin remodeling factors, non-coding RNA and signaling molecules have been implicated in impaired neural crest development that result in cardio-craniofacial syndromes (*Bélanger et al., 2018*; *Martik and Bronner, 2017*; *Plein et al., 2015*; *Singh et al., 2013*; *Strobl-Mazzulla et al., 2012*). However, the cell-autonomous role of splicing regulators in neural crest biology remains unclear and warrants further investigation.

AS of pre-messenger RNAs is essential for regulating gene expression and creating proteomic diversity. Changes in exon inclusion or exclusion can produce multiple mRNAs and protein isoforms with related or distinct functions. The RNA-binding fox-1 (Rbfox) homolog proteins: Rbfox1, Rbfox2, and Rbfox3 are evolutionarily conserved splicing factors that have been implicated in diverse cellular processes such as cell proliferation, cell death and epithelial-mesenchymal transition (*Kuroyanagi, 2009*). A conserved RNA-Recognition Motif that recognizes UGCAUG element in the introns, flanking target exons, characterizes Rbfox proteins (*Jin et al., 2003*). Rbfox proteins regulate splicing either positively or negatively, depending on their binding sites. They promote exon skipping or

inclusion when they bind to either the upstream or downstream of the alternative exon (*Jin et al., 2003*; *Ponthier et al., 2006*). In addition to splicing, Rbfox proteins also regulate transcriptional gene networks (*Fogel et al., 2012*). Although all Rbfox proteins bind to the same recognition sequence, their in vivo functions are only partially redundant due to their markedly distinct expression pattern. Both *Rbfox1* (*A2BP1*) and *Rbfox2* (*RBM9*) are expressed in the neurons, heart and skeletal muscle (*Gehman et al., 2012*; *Gehman et al., 2011*; *Singh et al., 2014*; *Wei et al., 2015*). Unlike the aforementioned, *Rbfox3* (*NeuN* or *HRNBP3*) expression is more restricted to neuronal tissues (*Kim et al., 2014*). Several diseases have been associated with dysregulation of splicing (*Garcia-Blanco et al., 2004*). For example, genetic deletion of *Rbfox1* and *Rbfox2* in the nervous system increased susceptibility to seizures and impaired cerebellum development, respectively (*Gehman et al., 2012*; *Gehman et al., 2011*). Morpholino-mediated knockdown of *Rbfox1* and *Rbfox2* in zebrafish embryos impaired cardiac and skeletal muscle functions (*Gallagher et al., 2011*). During skeletal muscle development, Rbfox2-mediated splicing is required for myoblast fusion and differentiation (*Runfola et al., 2015*; *Singh et al., 2014*). Decreased Rbfox2 expression has been reported in response to transverse aortic constriction in the mouse heart, and cardiac-specific deletion leads to pressure overload-induced heart failure (*Wei et al., 2015*). Abnormal expression of Rbfox2 has also been associated with hypoplastic left heart syndrome (*Verma et al., 2016*). Recently, *Nutter et al. (2016)* demonstrated that elevated Rbfox2 protein expression in diabetic hearts affects diabetes-induced AS of cardiac-specific genes. Despite extensive studies of Rbfox2 regulation on neuronal, cardiac and skeletal muscle cells, the cell-autonomous role of Rbfox2 in the functioning of neural crest cells has not been explored.

In the present study, we demonstrate that Rbfox2 is expressed in the neural crest cells (NCCs), neural crest-derived palate shelves, dorsal root ganglia, and somites. To determine the role of Rbfox2 during embryonic development, we deleted *Rbfox2* using floxed *Rbfox2* (*Rbfox2$^{flox/flox}$*) and *Pax3$^{Cre/+}$* knock in mice (*Engleka et al., 2005*; *Gehman et al., 2012*). We observed that *Pax3$^{Cre/+}$*-mediated deletion of *Rbfox2* results in neonatal lethality. *Rbfox2* mutant (*Rbfox2$^{Pax3-CKO}$*) embryos develop a cleft palate and defects in the development of craniofacial skeleton, suggesting defective cranial neural crest development. To determine whether cleft palate and craniofacial defects were due to deletion of *Rbfox2* in the neural crest cells, we generated a neural crest-specific *Rbfox2* mutant (*Rbfox2$^{Wnt1-CKO}$*) using *Wnt1$^{Cre/+}$* allele (*Gehman et al., 2012*; *Lewis et al., 2013*). Similar to *Rbfox2$^{Pax3-CKO}$* embryos, *Rbfox2$^{Wnt1-CKO}$* mutants developed a cleft palate and craniofacial skeleton defects and died postnatally at day 1. RNA-Seq analysis revealed an essential role of Rbfox2 in regulating splicing and expression of genes required for the development of cranial neural crest-derived structures. We demonstrate that Rbfox2-TGF-β-Tak1 signaling axis is impaired in the neural crest-derived cells of the *Rbfox2* mutant embryos and restoration of Tak1 expression in cultured palatal mesenchymal cells can rescue the proliferation defect observed in *Rbfox2* mutants. We also identified a positive *feedback loop* by which TGF-β signaling components, Smad2/3/4, bind directly to the Rbfox2 promoter and regulate its expression in the neural crest cells. Together, these results reveal a highly regulated Rbfox2-dependent splicing and transcriptional program that modulates cranial neural crest development.

## Results

### Rbfox2 is expressed in NCCs during mouse embryogenesis

To determine the expression pattern of Rbfox2, we performed Rbfox2 immunostaining on transverse sections from E9.5 to E11.5 embryos at different rostrocaudal axis. At E9.5, Rbfox2 is expressed in the premigratory NCCs at the dorsal neural tube, as well as in the migratory NCCs (*Figure 1A–C,F–H,K–M*) throughout the rostrocaudal axis, with strong expression of Rbfox2 observed in the somites. Rbfox2 expression is gradually reduced in migratory NCCs (*Figure 1K–M*). Rbfox2 expression was detected in the neural crest-derived craniofacial tissues including the palate shelves but not in the cardiac tissues such as OFT (*Figure 1D–E,I–J,N–O*). To compare Rbfox2 expression with a known neural crest cell marker, we performed Rbfox2 and Pax3 immunostaining on transverse sections from E9.5, E10.5 and E11.5 embryos (*Figure 1P–U*). At E9.5, Rbfox2 expression is identical to Pax3 in NCCs, dorsal neural tube, and somites (*Figure 1P and S*). At E10.5, in addition to premigratory NCCs, Rbfox2 is also expressed in the dorsal root ganglia (*Figure 1Q*). In contrast to Pax3

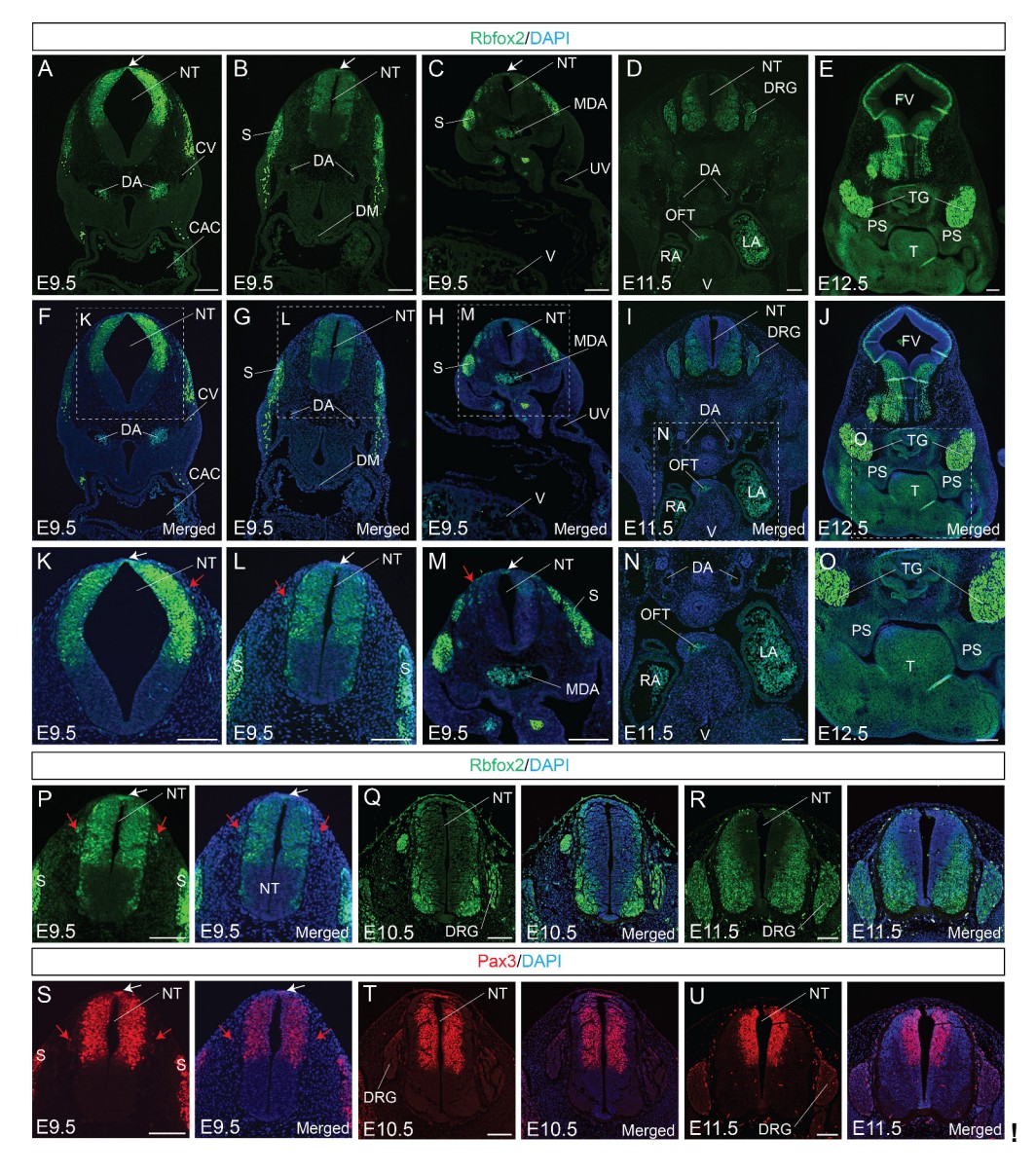

**Figure 1.** Rbfox2 is expressed in the neural crest cells during mouse embryonic development. Immunostaining for Rbfox2 was performed on E9.5, E10.5, E11.5 and E12.5 transverse sections at different rostrocaudal axis (**A–R**). Magnified view of the neural tube shows Rbfox2 expression in the pre-migratory (white arrows) and migratory (red arrows) neural crest cells (**K–M**). Rbfox2 expression in neural crest-derived tissues such as OFT and palate shelves (**D-E**, **I-J**, and **N-O**). Non-specific autofluorescence due to blood cells is observed in DA, MDA, OFT, RA and LA. Immunostaining for Rbfox2 and Pax3 was performed on adjacent sections from E9.5, E10.5, and E11.5 mouse embryos (**P–U**). Rbfox2 is expressed in the premigratory neural crest cells (white arrows) of the dorsal neural tube as well as in the migratory neural crest cells (red arrows). Nuclei were visualized by DAPI staining (blue). CAC, common atrial chamber and CV, cardinal vein; DGR, dorsal root ganglion; DM, dorsal mesocardium; DA, dorsal aorta; FV, fourth ventricle; LA, left atrium; MDA, midline dorsal aorta; NT, neural tube; OFT, outflow tract; PS, palatal shelves; RA, right atrium; S, somite; T, Tongue; TG, trigeminal ganglion; UV, umbilical vein; V, ventricle. Scale bars are 100 µm respectively.

DOI: https://doi.org/10.7554/eLife.45418.003

expression in the dorsal neural tube, the Rbfox2 expression is more restricted to the ventral neural tube (*Figure 1Q and T*). At E11.5, Rbfox2 is expressed in the ventral neural tube and dorsal root ganglia (*Figure 1R*). However, Pax3 is more restricted to the dorsal neural tube (*Figure 1U*). At an early stage, Rbfox2 expression was similar to that of Pax3, which is transiently expressed in all premigratory, migratory NCCs and somites. In contrast to Pax3 expression in the dorsal neural tube,

Rbfox2 expression was restricted to the ventral neural tube at later stages. These results indicate that Rbfox2 is expressed in NCCs, dorsal neural tube, palate shelves, dorsal root ganglia, and somites.

## *Rbfox2* deletion results in severe craniofacial defects, edema, and neonatal lethality

To establish the role of Rbfox2 during embryonic development, we conditionally deleted *Rbfox2* using *Pax3$^{Cre/+}$* knock in allele and floxed *Rbfox2* (*Rbfox2$^{flox/flox}$*) mice (*Engleka et al., 2005*; *Gehman et al., 2012*). *Pax3$^{Cre/+}$* knock in allele was used to delete *Rbfox2* not only in the premigratory and migratory NCCs, but also in paraxial mesoderm and somite derivatives where Rbfox2 is expressed. *Pax3$^{Cre/+}$;Rbfox2$^{flox/+}$* mice were fertile, born at the expected Mendelian ratio, and exhibited no gross abnormalities. However, we did not recover any mutant (*Rbfox2$^{Pax3-CKO}$*) pups at postnatal day 10 from breeding *Pax3$^{Cre/+}$;Rbfox2$^{flox/+}$* and *Rbfox2$^{flox/flox}$* mice, demonstrating that *Pax3$^{Cre/+}$*-mediated inactivation of *Rbfox2* leads to embryonic or neonatal lethality (*Figure 2—figure supplement 1A*). Genotyping of embryos from series of timed point matings demonstrated that *Rbfox2$^{Pax3-CKO}$* embryos are present at the expected Mendelian ratios at all embryonic time points analyzed (E10.5-E18.5) (*Figure 2—figure supplement 1A*). However, pups monitoring after birth revealed that mutant pups that are characterized by a shortened body axis and abnormal craniofacial features die at postnatal day one (P1) (*Figure 2—figure supplement 1B and G*). Our morphological and histological analyses demonstrate that *Rbfox2$^{Pax3-CKO}$* embryos develop cleft palate and severe subcutaneous edema, which are present in mid- and late-gestation (*Figure 2—figure supplement 1C–F,H–K*). Irregular breathing and hardly inflated lungs in *Rbfox2$^{Pax3-CKO}$* pups support the notion that the neonatal lethality is caused by respiratory failure (*Figure 2—figure supplement 1L–O*).

## Loss of *Rbfox2* in NCCs results in cleft palate

In *Rbfox2$^{Pax3-CKO}$* embryos, development of primary palates is not affected. However, all of *the Rbfox2$^{Pax3-CKO}$* pups died neonatally, exhibiting a secondary cleft palate defect. To determine when the cleft palate defects were first evident, we examined *Rbfox2$^{Pax3-CKO}$* embryos at progressively earlier stages by analyzing their morphological and histological data (*Figure 2A–L*). At E12.5, the palatal shelves in both control and *Rbfox2$^{Pax3-CKO}$* embryos were initiated normally, growing vertically flanking the developing tongue (*Figure 2A–B,G–H*). No obvious morphological or histological differences were observed at this embryonic stage. At E15.5 and E18.5, the control palatal shelves were elevated above the tongue to a horizontal position and met each other at the midline along the anterior-posterior axis (*Figure 2C–F*). However, in *Rbfox2$^{Pax3-CKO}$* embryos, palatal shelves were elevated above the tongue to a horizontal position, but they completely failed to fuse at the midline throughout the anterior-posterior axis (*Figure 2I–L*). Since *Pax3$^{Cre}$* is expressed in both the neural crest and mesodermal derivatives, it was not clear if the cleft palate defects were due to *Rbfox2* deletion in the neural crest cells. To determine the neural crest-specific requirement of *Rbfox2*, we generated neural crest-specific *Rbfox2* mutant (*Rbfox2$^{Wnt1-CKO}$*) embryos using *Wnt1$^{Cre/+}$* mice (*Gehman et al., 2012*; *Lewis et al., 2013*). Similar to *Rbfox2$^{Pax3-CKO}$*, *Rbfox2$^{Wnt1-CKO}$* mice were born at the expected Mendelian ratio but died at P1 (*Figure 2—figure supplement 1P–R*). To determine the craniofacial defects, we analyzed *Rbfox2$^{Wnt1-CKO}$* palates at different time points. Similar to *Rbfox2$^{Pax3-CKO}$* embryos, palatal shelves in *Rbfox2$^{Wnt1-CKO}$* embryos were elevated above the tongue to a horizontal position, but completely failed to fuse at the midline throughout the anterior-posterior axis (*Figure 2M–T*).

To determine the cellular mechanisms responsible for the impaired palatal growth in *Rbfox2* mutant embryos, we performed cell proliferation, and apoptosis assays on transverse palatal sections from control and *Rbfox2* mutant embryos. Ki67 immunohistochemistry revealed a significant reduction in cell proliferation in *Rbfox2* mutants when compared with control embryos at E12.5, in which the morphological or histological changes were not evident (*Figure 3A–C*). A significant difference in cell proliferation was more evident at E15.5 (*Figure 3D–F and G–I*). TUNEL assay demonstrated no differences in cell death between the control and *Rbfox2* mutants at E15.5 (*Figure 3J–K*). E-cadherin expression is observed in the nasal, palatal and tongue epithelium and not affected by *Rbfox2* deletion (*Figure 3L–M*). To determine whether cleft palate defect was due to impaired neural crest cell migration, we performed lineage-tracing analysis at E15.5 in both control (*Wnt1$^{Cre/+}$*:

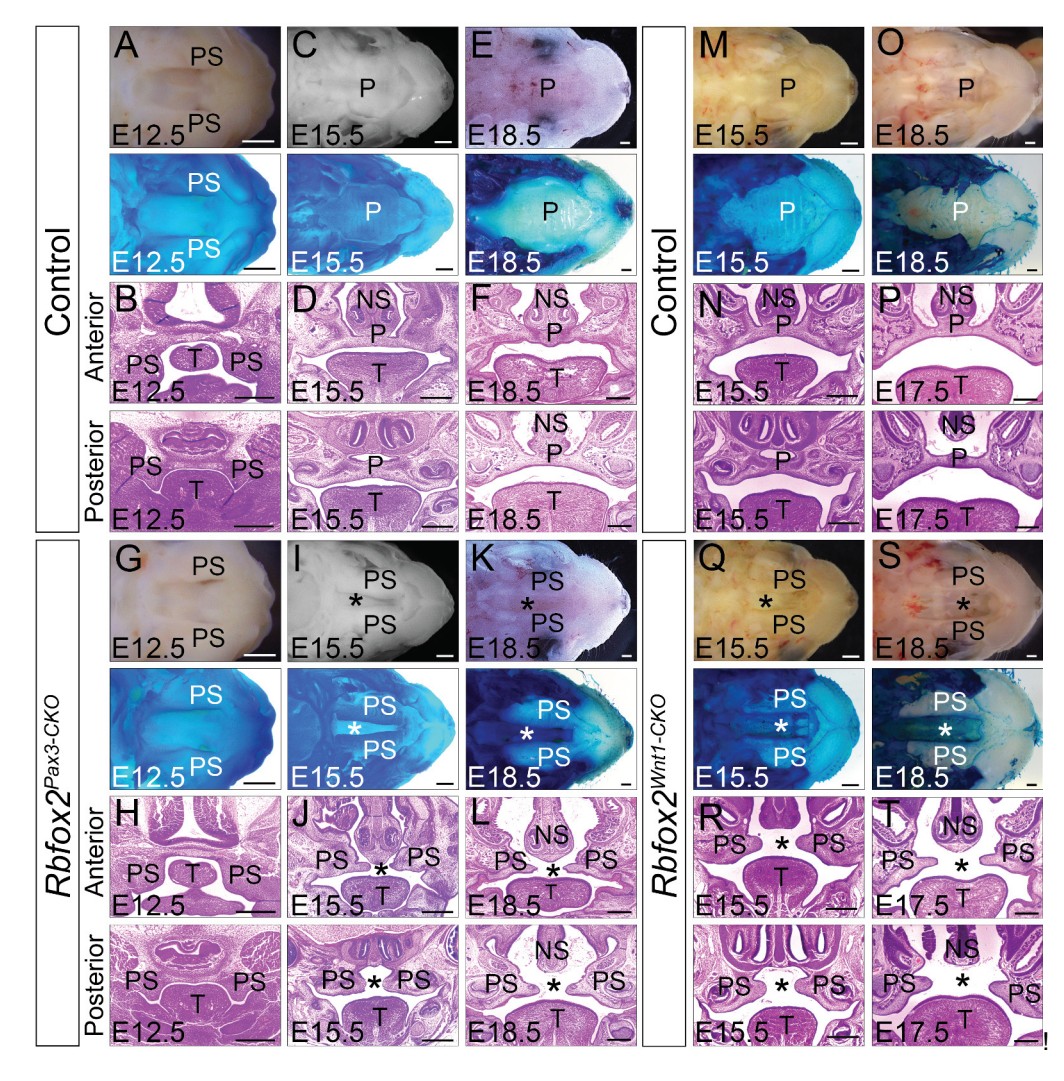

**Figure 2.** Cleft palate defects in *Rbfox2* mutant embryos. Gross morphology and Alcian blue staining of control (**A, C and E**) and *Rbfox2^Pax3-CKO^* (**G, I and K**) palates at E12.5 (n = 7 controls, n = 7 *Rbfox2^Pax3-CKO^*), E15.5 (n = 6 controls, n = 6 *Rbfox2^Pax3-CKO^*) and E18.5 (n = 7 controls, n = 7 *Rbfox2^Pax3-CKO^*). H and E stained transverse sections of control (**B**), (**D**), and (**F**) and *Rbfox2^Pax3-CKO^* (**H**), (**J**), and (**L**) embryos at the level of the anterior and posterior palatal shelves. Gross morphology and Alcian blue staining of control (**M and O**) and *Rbfox2^Wnt1-CKO^* (**Q and S**) palates at E15.5 (n = 6 controls, n = 6 *Rbfox2^Wnt1-CKO^*) and E18.5 (n = 7 controls, n = 6 *Rbfox2^Wnt1-CKO^*). H and E stained sections of control (**N and P**) and *Rbfox2^Wnt1-CKO^* (**R and T**) E15.5 and E17.5 embryos at the level of the anterior and posterior palatal shelves. Asterisks (*) represent cleft palate in *Rbfox2* mutant embryos (**I–L and Q–T**). P, palate; PS, palatal shelves; NS, Nasal septum; T, Tongue. Scale bars are 200 µm respectively.

DOI: https://doi.org/10.7554/eLife.45418.004

The following figure supplement is available for figure 2:

**Figure supplement 1.** *Rbfox2* deletion leads to neonatal lethality.

DOI: https://doi.org/10.7554/eLife.45418.005

*Rbfox2^flox/+^:R26^mTmG/+^*) and mutant (*Wnt1^Cre/+^:Rbfox2^flox/flox^:R26^mTmG/+^*) embryos. Labeled neural crest cells marked by GFP immunostaining were abundantly present in the palate shelves of mutant embryos (**Figure 3N–O**). To determine any gross abnormality in NCCs migration, we performed line-age-tracing analysis at E12.5 and E14.5 in both control (*Pax3^Cre/+^:Rbfox2^flox/+^:R26^mTmG/+^*) and mutant (*Pax3^Cre/+^:Rbfox2^flox/flox^:R26^mTmG/+^*) embryos. No obvious NCCs migration defect was observed (**Figure 3—figure supplement 1**).

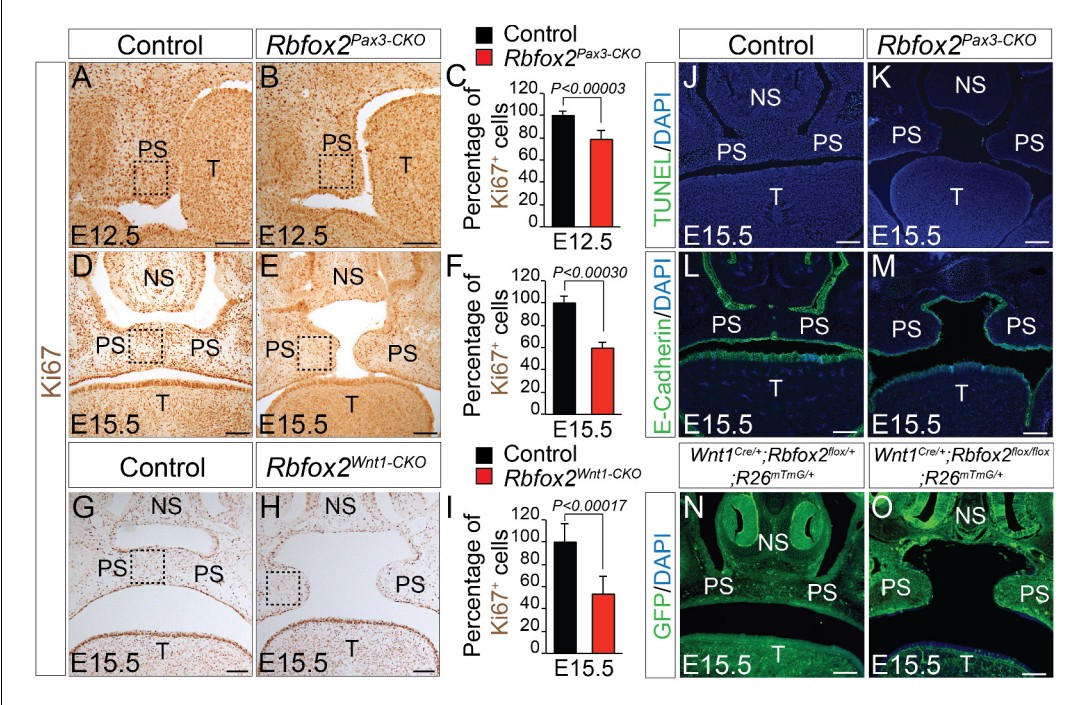

**Figure 3.** Cleft palate defects in *Rbfox2* mutant embryos result from impaired cell proliferation in the neural crest-derived palatal shelves. Immunohistochemistry for Ki67 was performed on transverse sections through the middle palatal regions of E12.5 (n = 4 controls, n = 4 *Rbfox2^Pax3-CKO^*) and E15.5 (n = 5 controls, n = 5 *Rbfox2^Pax3-CKO^*) control (**A, D**) and *Rbfox2^Pax3-CKO^* (**B, E**) embryos. Immunohistochemistry for Ki67 was performed on E15.5 control (**G**) and *Rbfox2^Wnt1-CKO^* (**H**) middle palatal shelves sections (n = 5 controls, n = 5 *Rbfox2^Wnt1-CKO^*). Quantification of cell proliferation was calculated as the ratio of Ki67-positive cells to the total number of cells as determined by DAPI counterstain in the defined area of palatal shelves (**C, F** and **I**). TUNEL assay was performed on E15.5 control (**J**) and *Rbfox2^Pax3-CKO^* (**K**) sections (n = 4 controls, n = 4 *Rbfox2^Pax3-CKO^*). E-cadherin immunostaining on E15.5 control (**L**) and *Rbfox2^Pax3-CKO^* (**M**) sections (n = 4 controls, n = 4 *Rbfox2^Pax3-CKO^*). GFP immunostaining on E15.5 *Wnt1^Cre/+^*; *Rbfox2^flox/+^*;*R26^mTmG/+^* (**N**) and *Wnt1^Cre/+^*;*Rbfox2^flox/flox^*;*R26^mTmG/+^* (**O**) sections showing neural crest derivatives cells in the palatal shelves (n = 4 each genotype). NS, Nasal septum; PS, palatal shelves; T, Tongue. Scale bars are 100 μm respectively.

DOI: https://doi.org/10.7554/eLife.45418.006

The following figure supplement is available for figure 3:

**Figure supplement 1.** Neural crest cell migration is not affected in *Rbfox2* mutant embryos.

DOI: https://doi.org/10.7554/eLife.45418.007

## Craniofacial and axial skeletal defects in *Rbfox2* mutants

To examine the nature and severity of the skeleton defects, we performed Alizarin Red S and Alcian Blue staining on *Rbfox2^Pax3-CKO^* and *Rbfox2^Wnt1-CKO^* embryos (**Figure 4**). In both *Rbfox2^Pax3-CKO^* and *Rbfox2^Wnt1-CKO^* embryos, neural crest-derived bones such as frontal bones of the calvaria were hypoplastic and widely separated leaving a wide dorsal opening (**Figure 4C and M**). Decreased ossification of nasal bone was observed in *Rbfox2^Pax3-CKO^* mutant embryos compared to the controls (**Figure 4B–C and L–M**). Reduction in the lower jaw or mandible size was observed in *Rbfox2* mutant embryos (**Figure 4E–H and N–Q**). Further analysis revealed that both the shape and size of most neural crest-derived bones including alisphenoid, premaxilla, palatal process of premaxilla, palatal process of maxilla and palatine are affected in *Rbfox2* mutant embryos (**Figure 4J–K and R–S**). The palatal process of palatine bone is also missing in *Rbfox2* mutant embryos (**Figure 4K and S**).

The whole embryo skeletal preparations displayed severe defects in the axial skeleton of *Rbfox2-^Pax3-CKO^* embryos (**Figure 4—figure supplement 1A**). Mutants are characterized by their shortened axial skeleton and smaller thoracic cavity. The vertebral column of control embryos showed a clear S-bend in the cervical and thoracic region (**Figure 4—figure supplement 1A**). However, in the *Rbfox2^Pax3-CKO^* embryos, the vertebral column was rather straight, positioning the skull and vertebral column perpendicular to each other (**Figure 4—figure supplement 1A**). Ectopic bone formation and fusion of vertebral bodies were observed in *Rbfox2^Pax3-CKO^* embryos (**Figure 4—figure**

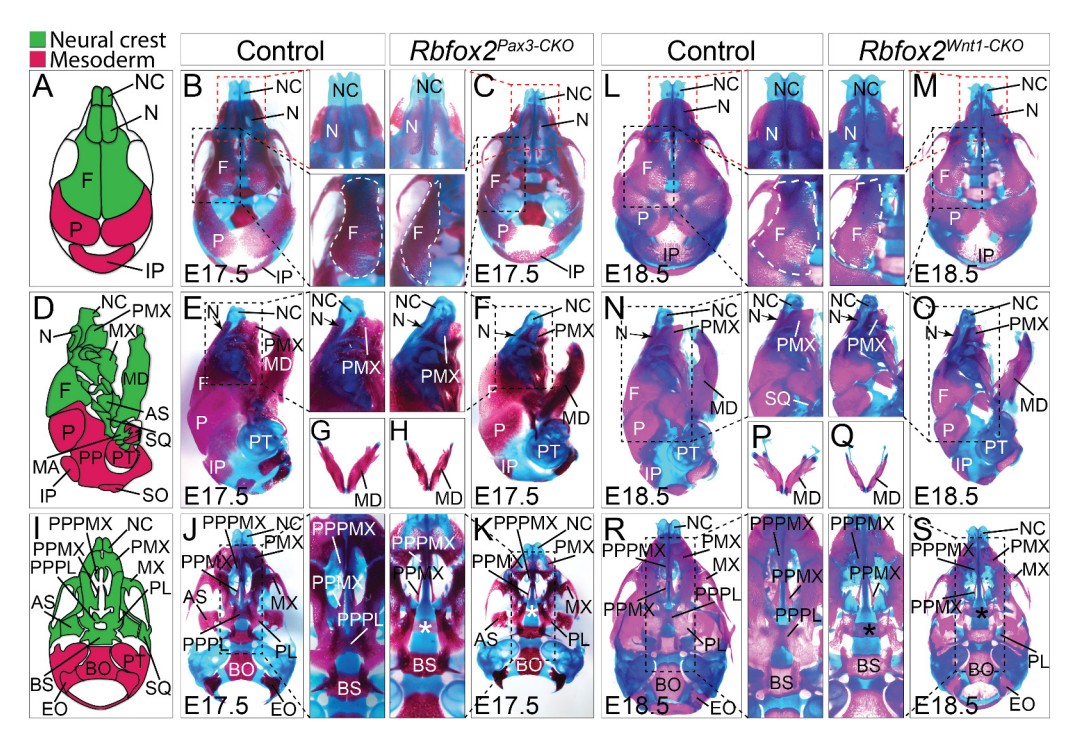

**Figure 4.** Craniofacial skeleton defects in *Rbfox2* mutant embryos. Alizarin red and Alcian blue stainings for ossified and chondrified tissues, respectively, of control (A–B), (D–E), (G), and I–J) and *Rbfox2^Pax3-CKO^* (C), (F), (H), and (K) skeleton at E17.5 (n = 8 controls, n = 6 *Rbfox2^Pax3-CKO^*). Alizarin red and Alcian blue stainings of control (L), (N), (P), and (R) and *Rbfox2^Wnt1-CKO^* (M, O, Q and S) skeleton at E18.5 (n = 8 controls, n = 5 *Rbfox2^Wnt1-CKO^*). Neural crest and mesoderm contribution to craniofacial bones are represented in green and pink color respectively (A, D and I). Dorsal (A–C and L–M), lateral (D–F and N–O) and ventral (I–K and R–S) view of skulls. Both *Rbfox2^Pax3-CKO^* (C), (F), (H), and K) and *Rbfox2^Wnt1-CKO^* (M), (O), (Q), and S) embryos demonstrate severe hypoplasia and diminished ossification of many neural crest-derived bones. Asterisks (*) represent the missing PPPL bone in *Rbfox2* mutant embryos (K and S). AS, alisphenoid; BO, basioccipital; BS, basisphenoid; EO, exoccipital; F, frontal bone; IP, interparietal; MD, mandible; MX, maxilla; N, nasal; NC, nasal capsule; P, parietal bone; PL, palatine; PMX, premaxilla; PPMX, palatal process of maxilla; PPPL, palatal process of palatine; PPPMX, palatal process of premaxilla; PT, petrous part of temporal bone; SO, supraoccipital; SQ, squamous.

DOI: https://doi.org/10.7554/eLife.45418.008

The following figure supplements are available for figure 4:

**Figure supplement 1.** Skeletal defects in *Rbfox2* mutant embryos.
DOI: https://doi.org/10.7554/eLife.45418.009

**Figure supplement 2.** Neural crest contribution to the cardiac OFT is not affected in *Rbfox2* knockout embryos.
DOI: https://doi.org/10.7554/eLife.45418.010

**Figure supplement 3.** Development of peripheral and enteric nervous system in *Rbfox2* mutant embryos.
DOI: https://doi.org/10.7554/eLife.45418.011

**Figure supplement 4.** Thymus development is grossly intact in *Rbfox2* mutant embryos.
DOI: https://doi.org/10.7554/eLife.45418.012

**Figure supplement 5.** Neural crest contribution to adrenal gland is not affected in *Rbfox2* knockout embryos.
DOI: https://doi.org/10.7554/eLife.45418.013

**Figure supplement 6.** Limb and diaphragm musculature are not affected in *Rbfox2^Pax3-CKO^* embryos.
DOI: https://doi.org/10.7554/eLife.45418.014

supplement 1A). No defects in the axial skeletons were observed in *Rbfox2^Wnt1-CKO^* embryos (*Figure 4—figure supplement 1B*). Von Kossa staining of the calvaria from E17.5 *Rbfox2^Pax3-CKO^* mutant embryos revealed the impaired development of mesenchymal condensations that become ossified bone (*Figure 4—figure supplement 1C*). Reduced thickness in the ossified calvaria bone was observed in *Rbfox2^Pax3-CKO^* embryos.

## *Rbfox2* deletion does not affect other neural crest-derived structures except cranial nerves

As shown by normal septation and alignment of the aorta and pulmonary trunk, the development of cardiac OFT was not affected in *Rbfox2* mutants (*Figure 4—figure supplement 2A–J*). No change in smooth muscle actin (SMA) staining was observed (*Figure 4—figure supplement 2C,F*). Fate-mapping analysis demonstrated that cardiac NCCs migration was grossly intact in the *Rbfox2* mutant embryos (*Figure 4—figure supplement 2K–P*). To determine the effect of *Rbfox2* deletion on neurons that populate the dorsal root, sympathetic and enteric ganglia, we performed whole mount neurofilament (2H3) staining to mark the differentiated neurons. The neurofilament 2H3 staining in *Rbfox2* mutant embryos revealed abnormalities in the oculomotor (III), trochlear (IV) and hypoglossal nerve (XII cranial nerve) (*Figure 4—figure supplement 3A*). Oculomotor and trochlear nerves appear to be deformed. Analysis of cranial ganglia at higher magnification revealed that the roots of the hypoglossal cranial nerve are not fully developed. The hypoglossal cranial nerve is disorganized and shorter in *Rbfox2*[Pax3-CKO] embryos (*Figure 4—figure supplement 3A*). Hypoglossal nerve defects could be secondary to defects in the hypoglossal cord which is derived from the occipital somites, and where Pax3 is expressed (*Bajard et al., 2006*). No significant difference was observed in the size of the dorsal root ganglion (*Figure 4—figure supplement 3B*). Enteric nervous system development was not affected in the absence of *Rbfox2* (*Figure 4—figure supplement 3C–D*). Similarly, other neural crest-derived organs such as thymus and adrenal gland (chromaffin cells) were not affected in *Rbfox2* mutant embryos (*Figure 4—figure supplement 4*, and *Figure 4—figure supplement 5*). Since *Pax3*[Cre/+] transgene is active in non-neural crest-derived tissues such as limb and diaphragm muscles, we also analyzed these tissues and found no significant changes (*Figure 4—figure supplement 6*).

## Splicing and transcriptional changes in cranial neural crest-derived structures in response to *Rbfox2* deletion

To determine the splicing program and transcriptional network regulated by Rbfox2 in vivo, we performed RNA-Seq profiling using poly(A)[+] RNA isolated from microdissected craniofacial tissues from E12.5 control and *Rbfox2* mutant embryos (*Figure 5A*). We performed RNA-Seq analysis at E12.5 because of the minimal morphological and structural changes observed in *Rbfox2* mutant embryos at this stage. Multiplexed libraries were prepared for all replicates and sequenced together using Illumina HiSeq 4000 platform to produce 65–80 million, 151-nucleotide paired-end reads per sample (see Materials and methods for detail). Paired-end fastq sequence reads from each sample were aligned to mouse reference genome using ultrafast RNA-Seq aligner STAR with 82% average mapping rate and negligible ribosomal RNA contamination (<1%). Differential expression of genes and transcript isoforms between controls and *Rbfox2* mutant samples were determined using two tools: MISO (Mixture of Isoforms), which quantitates the expression level of alternatively spliced genes and identifies differentially regulated isoforms or exons across samples and Cuffdiff, which finds significant changes in transcript expression, splicing, and promoter use. Using MISO analysis, we identified 81 differentially expressed transcripts from 59 genes in *Rbfox2* mutant samples as compared with control (*Figure 5B* and *Figure 5—source data 1*). However, Cuffdiff analysis identified 33 alternatively spliced transcripts from 30 genes in *Rbfox2* mutant samples as compared with controls (*Figure 5C* and *Figure 5—source data 2*).

Pathway enrichment analysis identified significant enrichment of genes that control cellular and anatomical structure morphogenesis (*Figure 5D*). We analyzed the location of UGCAUG sequences in different gene features (5'-UTR, promoter, exons, introns, 3-UTR) of Rbfox2 target genes identified through Cuffdiff and MISO analysis. We performed an analysis of motif enrichment (AME) through the MEME suite (version 5.0.5) and identified significant UGCAUG motif enrichment only in the introns of the Rbfox2 target genes (adjusted p-value=1.4E-08) (*Figure 5E*). We also performed AME on 75 randomly selected genes whose expression levels were comparable to the Rbfox2-target genes, but did not show any transcript-level differential expression in response to *Rbfox2* deletion. However, no significant motif enrichment was observed (adjusted p-value=1.0E-01). Next, we used the FIMO tool in MEME to analyze the location of intronic UGCAUG sequences in the Rbfox2 target genes. Most UGCAUG sequences were located within 1000 bases of the exon on either flanking introns (*Figure 5F*). Venn analysis was performed to identify a small set of high-confidence

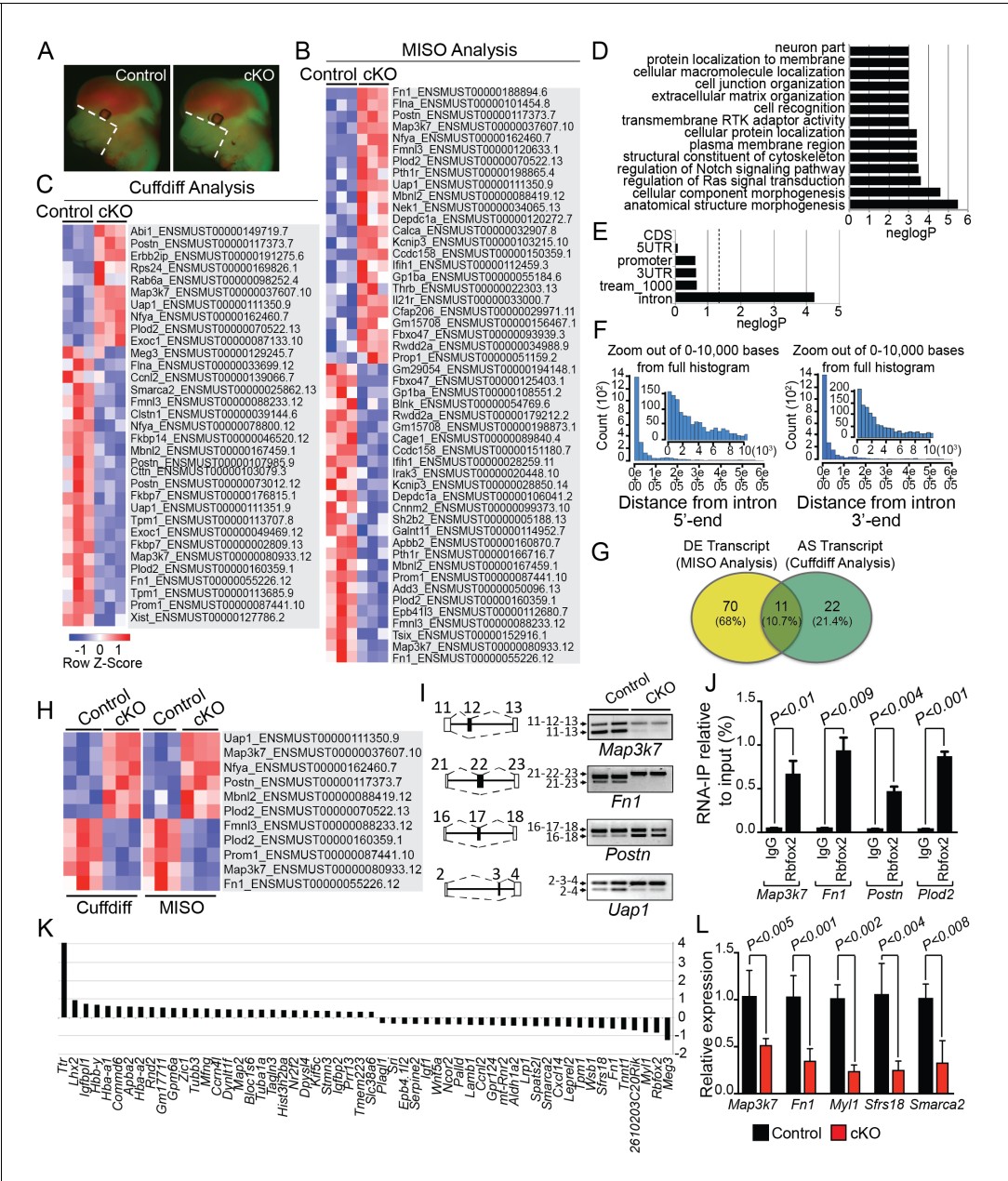

**Figure 5.** Rbfox2-dependent splicing and transcriptional changes in neural crest cells. Representative fluorescent images from E12.5 control (n = 3) and *Rbfox2^Pax3-CKO* mutant (represented as cKO, n = 3) embryos showing the area (white, dotted line) of craniofacial tissue microdissected for RNA-Seq analysis (A). Heat map of 50 differentially expressed transcripts (out of a total of 81 transcripts) identified by MISO analysis of RNA-Seq data (B). Heat map of 33 alternatively spliced transcripts identified by Cuffdiff analysis of RNA-Seq data (C). Pathway enrichment analysis of Rbfox2 target genes (D). Motif (UGCAUG) enrichment analysis. Significant enrichment was observed only in the introns of the target genes (E). Location of the intronic UGCAUG sequences in the Rbfox2 target genes (F). Venn diagram showing the overlap of transcripts identified by MISO and Cuffdiff analysis of RNA-Seq data (G). Heat map representation of 11 transcripts identified by both MISO and Cuffdiff analysis (H). RT-PCR analysis of alternative splicing of Rbfox2 targets in control and *Rbfox2* mutant cranial mesenchyme from two independent replicates for each genotype (I). The gene structure, illustrating the alternative exons is presented on left. RNA-IP on palatal mesenchymal cells (J). Differentially expressed genes control and *Rbfox2* mutants (K). qRT-PCR validation of differentially spliced and differentially expressed genes (L).

DOI: https://doi.org/10.7554/eLife.45418.015

The following source data and figure supplement are available for figure 5:

**Source data 1.** A complete list of transcripts identified by MISO analysis.
DOI: https://doi.org/10.7554/eLife.45418.017

**Source data 2.** A complete list of transcripts identified by Cuffdiff analysis.

*Figure 5 continued on next page*

*Figure 5 continued*

DOI: https://doi.org/10.7554/eLife.45418.018
**Figure supplement 1.** Changes in alternative splicing of Rbfox2 target genes.
DOI: https://doi.org/10.7554/eLife.45418.016

alternatively spliced transcripts that could be validated in downstream experiments. Venn analysis identified 11 transcripts that were identified by both programs and constituted a high confidence set of differentially expressed and AS isoforms in *Rbfox2* mutants (*Figure 5G–H* and *Figure 5—figure supplement 1*). Specific genes identified by Venn analysis were selected for validation by RT-PCR based on their established function in the neural crest or craniofacial development. Using RNA isolated from control and *Rbfox2* knockout craniofacial tissues, we validated the splicing changes identified by RNA-Seq by performing reverse transcriptase PCR (RT-PCR) in a selected group of genes such as mitogen-activated protein kinase kinase kinase 7 (*Map3k7*), fibronectin 1 (*Fn1*), periostin (*Postn*) and UAP1 UDP-N-acetylglucosamine pyrophosphorylase 1 (*Uap1*). *Rbfox2* deletion significantly impacted splicing and expression of these candidate genes (*Figure 5I–L*). For example, *Rbfox2* deletion reduced the expression of the predominantly expressed *Map3k7* transcript. The short *Fn1* transcript was not present in *Rbfox2* mutants. To determine that Rbfox2 directly bind to these target genes in vivo, we performed RNA immunoprecipitation (RIP) assays using anti-Rbfox2 antibody on palatal mesenchymal cells, followed by quantitative RT-PCR. We observed significant enrichment suggesting that Rbfox2 can directly bind to the RNA of these target genes and modulate splicing (*Figure 5J*). To determine the transcriptional changes associated with the craniofacial defects, we analyzed genes that are differentially expressed between control and *Rbfox2* knockouts. We identified 56 differentially expressed genes (*Figure 5K*). We further validated the expression of candidate genes that were either differentially expressed or spliced by quantitative PCR in both control and *Rbfox2* knockout tissues (*Figure 5L*). We observed a significant reduction in *Map3k7*, *Fn1*, *Myl1*, *Sfrs18* and *Smarca2* in *Rbfox2* knockout tissues (*Figure 5L*). Together, we identified over 100 AS transcripts and 56 genes that are differentially expressed between control and *Rbfox2* mutant embryos.

## Impaired Rbfox2-TGFβ-Tak1 signaling axis in *Rbfox2* mutant embryos

By analyzing alternatively spliced and differentially expressed genes from the RNA-Seq data, we observed that a number of genes known to affect TGF-β signaling pathway, such as *Map3k7*, *Fn1*, *Meg3* etc. were significantly reduced in *Rbfox2* mutants. This led us to hypothesize that TGF-β signaling pathway may be affected by *Rbfox2* deletion. TGF-β signaling pathway involves both canonical and non-canonical signaling cascades. Recent work has shown that Tak1, encoded by *Map3k7* is required for activation of both canonical and non-canonical signaling. To identify if TGF-β signaling pathway is affected by *Rbfox2* deletion in vivo, we harvested palatal shelves from control and *Rbfox2* mutant embryos and performed western blot analysis for Tak1 and phosphorylated Tak1. We found that both Tak1 and phosphorylated Tak1 levels were significantly reduced in *Rbfox2* mutant embryos, demonstrating impaired TGF-β signaling pathway in neural crest-derived palate shelves (*Figure 6A*). We further investigated the downstream signaling targets of Tak1 such as p38 Mapk, phosphorylated p38 Mapk, Smad2, and phosphorylated Smad2. The level of C-terminal Smad2 phosphorylation was significantly reduced; however, no change in total Smad2 was observed. Similarly, the level of phosphorylated p38 Mapk was significantly reduced, although there was no change in total p38 Mapk (*Figure 6A* and *Figure 6—figure supplement 1*). Consistent with the reduction in its mRNA levels, Fn1 protein levels were also significantly reduced in *Rbfox2* mutant neural crest as compared with controls (*Figure 6B* and *Figure 6—figure supplement 1*). Next, we tested how neural crest-derived palatal mesenchymal cells respond to TGF-β stimulation. We established culture conditions to grow palatal mesenchymal cells from control and *Rbfox2* mutant embryos. The neural crest cell's origin and purity were confirmed by growing cultures of *Pax3$^{Cre/+}$;Rbfox2$^{flox/+}$; Rosa26$^{mTmG/+}$* embryos. Majority of the cultured cells were GFP positive confirming their neural crest origin (*Figure 6C*). Consistent with the in vivo data, palatal mesenchymal cells isolated from *Rbfox2* mutant embryos showed reduced levels of both Tak1 and phosphorylated Tak1 when compared with control following TGF-β stimulation (*Figure 6D*). In addition, no significant change in pTak1/

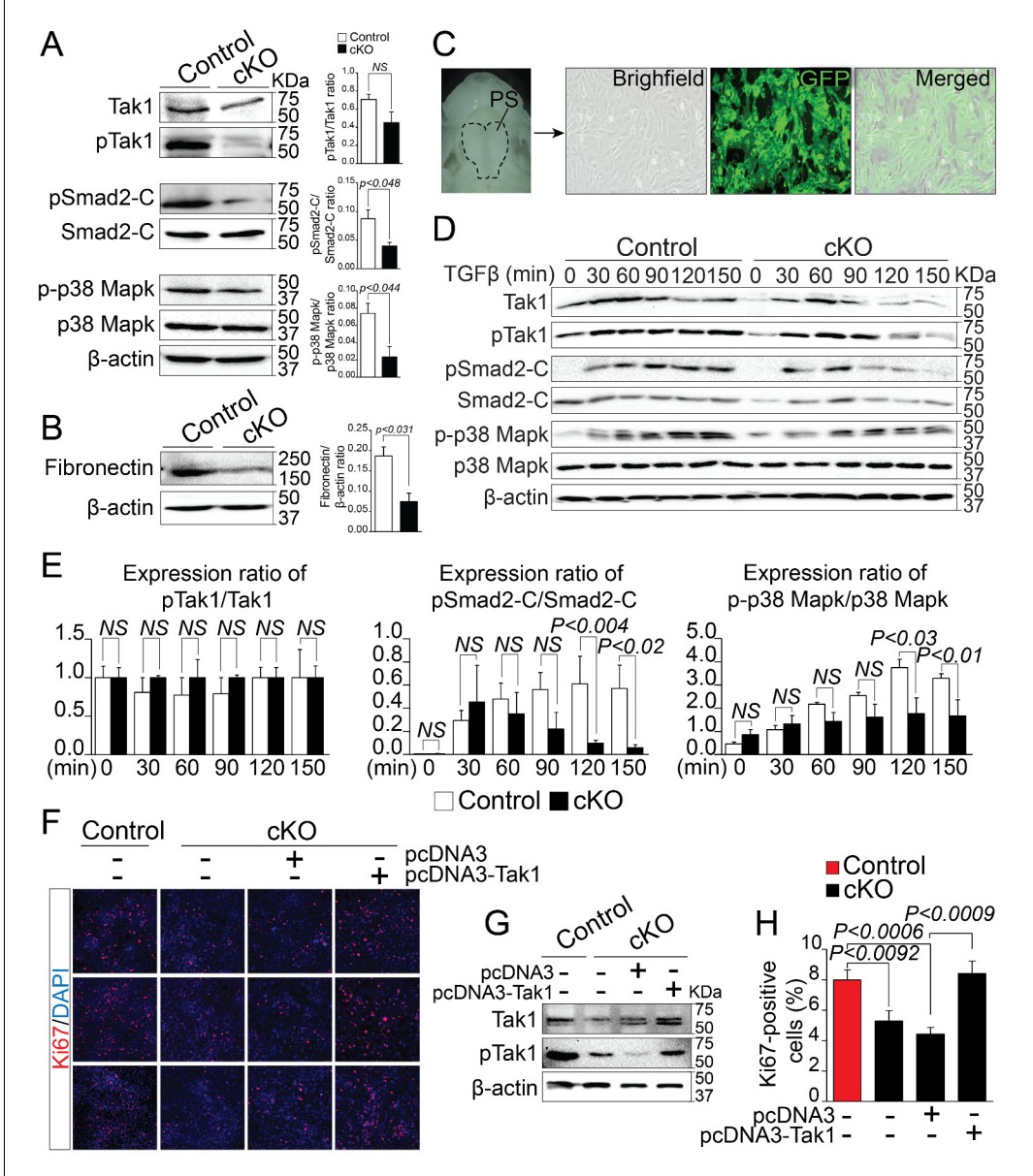

**Figure 6.** Deregulation of Rbfox2-TGFβ-Tak1 signaling axis in *Rbfox2* mutant embryos. Representative western blot and quantification from microdissected palatal shelves from control and *Rbfox2^Pax3-CKO* embryos (represented as cKO) at E14.5 and analyzed for Tak1, phospho-Tak1, Smad2, phospho-Smad2-C, p38 Mapk and phospho- p38 Mapk (**A**). Western blot and quantification for fibronectin on control and *Rbfox2* mutant embryos at E14.5 (**B**). Primary palatal mesenchymal cell cultures were established from E14.5 *Pax3^Cre/+*; *Rbfox2^flox/+*; *R26^mTmG/+* embryos. Representative bright field and fluorescent images were taken. The majority of the cultured cells are GFP positive demonstrating their neural crest origin (**C**). Representative western blot and quantification from control and *Rbfox2* mutant primary palatal mesenchymal cells stimulated with recombinant TGFβ and analyzed for Tak1, phospho-Tak1, Smad2, phospho-Smad2-C, p38 Mapk and phospho-p38 Mapk (**D**). Quantification of western blot (**E**). Overexpression of Tak1 in *Rbfox2* mutant palatal mesenchymal cells. Ki67 immunostaining to determine palatal mesenchymal cell proliferation (**F**). Representative western blot for Tak1 and pTak1 (**G**). Quantification of the percentage of Ki67 positive cells (**H**). β-Actin was used as loading control. PS, palatal shelves; *NS*, not significant.

DOI: https://doi.org/10.7554/eLife.45418.019

The following figure supplements are available for figure 6:

**Figure supplement 1.** Impaired Rbfox2-TGFβ-Tak1 signaling axis due to *Rbfox2* deletion.
DOI: https://doi.org/10.7554/eLife.45418.020
**Figure supplement 2.** Effect of Tak1 overexpression on Rbfox2 target genes.
DOI: https://doi.org/10.7554/eLife.45418.021

Tak1 ratio was observed, suggesting that Tak1 activation was not affected by *Rbfox2* deletion (*Figure 6E*). Similarly, TGF-β-induced C-terminal Smad2 and p38 Mapk phosphorylation levels were reduced in *Rbfox2* mutant cells when compared with controls, while no significant change was observed in total Smad2 or p38 Mapk (*Figure 6D–E* and *Figure 6—figure supplement 1*). Quantification of these proteins showed a significant reduction in the ratio of pSmad2-C/Smad2-C and p-p38 Mapk/p38 Mapk, suggesting that Rbfox2-TGF-β-Tak1 signaling axis was impaired in *Rbfox2* mutant cells (*Figure 6D–E* and *Figure 6—figure supplement 1*).

To determine whether Tak1 overexpression can restore TGF-β signaling pathway and rescue the proliferation defects seen in *Rbfox2* mutant embryos in vivo, we cultured palatal mesenchymal cells from control and *Rbfox2* mutant embryos. We then transfected *Rbfox2* mutant cells with either empty plasmid (pcDNA3) or plasmid expressing Tak1 (pcDNA3-Tak1), and performed Ki67 immunostaining (*Figure 6F–H*). We confirmed that both Tak1 and phosphorylated Tak1 levels were significantly increased after Tak1 transfection in *Rbfox2* mutant cells (*Figure 6G*). Consistent with the in vivo data, *Rbfox2* mutant palatal cells proliferate slower than control cells and Tak1 overexpression can rescue the proliferation defects (*Figure 6H*). To determine the effect of Tak1 overexpression on the expression of Rbfox2-dependent genes, we performed qRT-PCR for *Map3k7*, *Fn1*, *Myl1*, *Sfrs18* and *Smarca2* on control and *Rbfox2* mutant cells transfected with either empty plasmid vector or plasmid expressing Tak1 (*Figure 6—figure supplement 2*). Compared with empty vector transfected controls, we observed significant increase in the expression of *Map3k7* and *Fn1* in Tak1 overexpressing *Rbfox2* mutant cells. No change in *Myl1*, *Sfrs18* and *Smarca2* expression was observed (*Figure 6—figure supplement 2*). Together, these results demonstrate that *Rbfox2* deficiency leads to impaired Rbfox2-TGF-β-Tak1 signaling axis, resulting in reduced palatal cell proliferation. In addition, Tak1 overexpression restores TGF-β signaling pathway and rescues the proliferation defects in *Rbfox2* mutant cells.

## Rbfox2 expression is regulated by TGF-β signaling pathway in a feedback loop

To determine whether TGF-β signaling pathway regulate Rbfox2 expression in a positive feedback loop, we analyzed Rbfox2 expression in wildtype palatal mesenchymal cells after TGF-β stimulation. We observed a significant increase in Rbfox2 expression after TGF-β treatment (*Figure 7A* and *Figure 7—figure supplement 1*). To determine the mechanism by which TGF-β regulates Rbfox2 expression, we stimulated palatal mesenchymal cells with TGF-β in the presence or absence of chemical inhibitors blocking either canonical (Smad3 inhibitor) or non-canonical (Tak1 and p38 inhibitor) TGF-β signaling pathway (*Figure 7B–D* and *Figure 7—figure supplement 1*). We observed that Smad3 inhibitor abolished TGF-β-induced increase in Rbfox2 expression (*Figure 7B* and *Figure 7—figure supplement 1*). A similar trend was observed with Tak1 inhibitor (*Figure 7C* and *Figure 7—figure supplement 1*). However, p38 inhibitor did not impact TGF-β-induced Rbfox2 expression (*Figure 7D* and *Figure 7—figure supplement 1*). Together, these results suggest that TGF-β induces Rbfox2 expression either through Tak1-dependent or -independent canonical pathways. To further investigate how Smad-dependent canonical pathway regulates Rbfox2 expression, we analyzed 2.5 kb *Rbfox2* promoter and identified two sites with multiple Smad binding elements (SBEs). *Rbfox2* promoter fragment (1.6 kb) with multiple SBEs was PCR-amplified, cloned into a luciferase reporter plasmid, and tested in luciferase reporter assays (*Figure 7—figure supplement 2*). Smad2, Smad3 or Smad4 significantly activated the Rbfox2 luciferase reporter in the presence/absence of recombinant TGF-β. However, in the presence of recombinant TGF-β, the fold activation was much higher when compared with no TGF-β stimulation (*Figure 7E–G*). Using the wild-type palatal mesenchymal cells in the presence or absence of TGF-β, we next tested Smad2/3 binding activity to SBEs identified in the *Rbfox2* promoter in vivo by chromatin immunoprecipitation (ChIP) assays. Our data indicate that Smad2/3 binds directly to these sites in vivo. Moreover, we observed enrichment in Smad2/3 chromatin binding after TGF-β treatment (*Figure 7H*). Together, these results demonstrate that *Rbfox2* expression in neural crest-derived palatal mesenchymal cells is tightly regulated by TGF-β signaling pathway (*Figure 7I*).

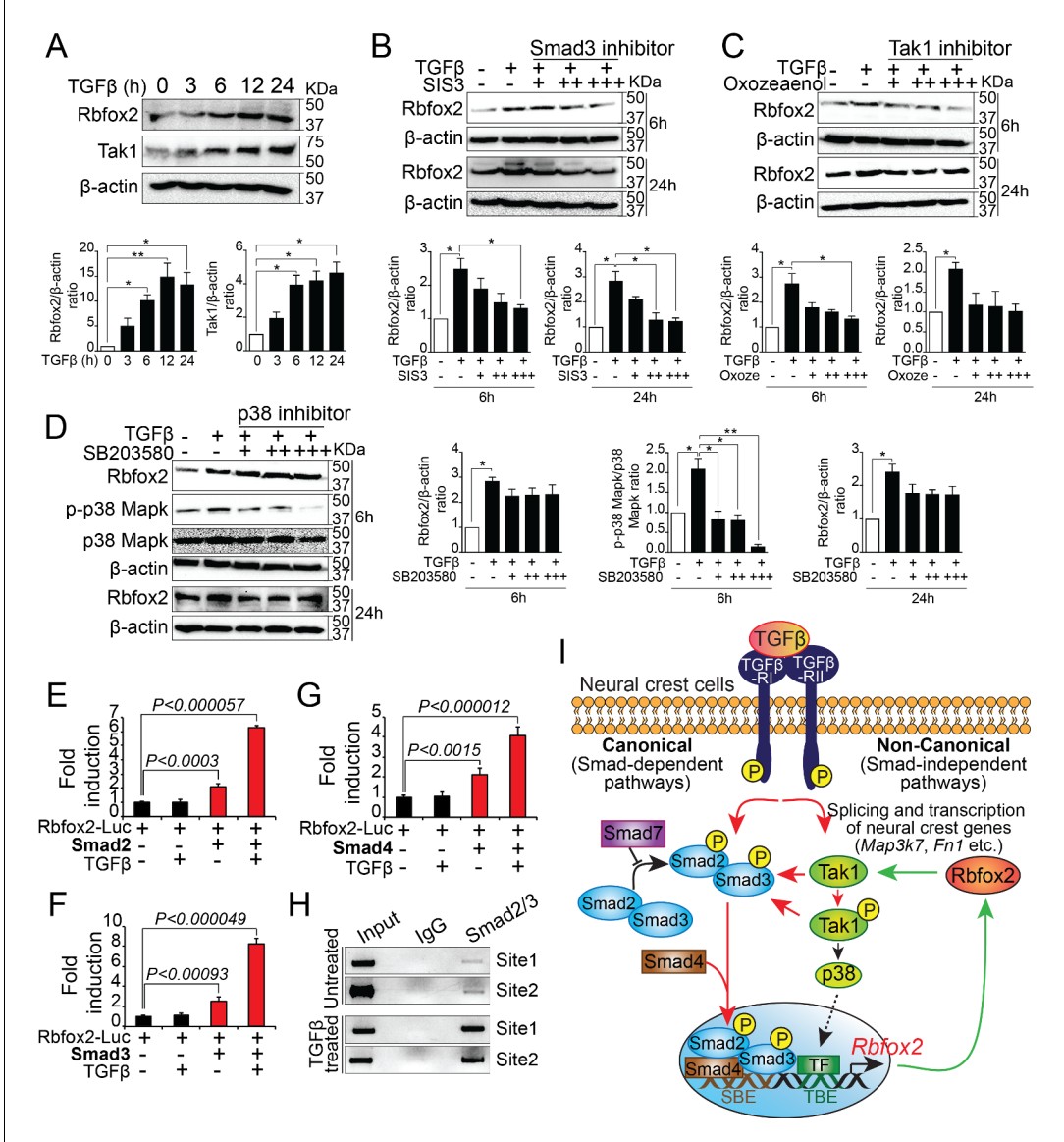

**Figure 7.** Rbfox2 expression is regulated by TGFβ signaling pathway in neural crest-derived palatal cells. Representative western blot and quantification demonstrating induction of Rbfox2 in primary palatal mesenchymal cells stimulated with recombinant TGFβ. β-Actin was used as a loading control (**A**). Representative Rbfox2 western blot and quantification from primary palatal mesenchymal cells stimulated with recombinant TGFβ with and without, Smad3 inhibitor (SIS3) or Tak1 inhibitor (5Z-7-Oxozeaenol) or p38 inhibitor (SB203580) at 6 and 24 hr time points. β-Actin was used as a loading control (**B–D**). *, p<0.05; **, p<0.01 (one-way ANOVA between groups, Bonferroni's multiple comparisons test). The *Rbfox2*-luciferase reporter was transfected in HEK293T cells with or without recombinant TGFβ in the presence or absence of Smad2 (**E**) or Smad3 (**F**) or Smad4 (**G**). ChIP assay using chromatin from untreated or TGFβ treated primary palatal mesenchymal cells and Smad2/3 antibody. Predicted binding sites in *Rbfox2* promoter were tested (**H**). Model depicting the interaction between the TGF-β signaling pathway and alternative splicing factor Rbfox2 in neural crest cells. Rbfox2 regulates alternative splicing and transcription of neural crest genes. In a feedback loop, Rbfox2 expression is regulated by the canonical TGFβ signaling pathway (red arrows) (**I**).

DOI: https://doi.org/10.7554/eLife.45418.022

The following figure supplements are available for figure 7:

**Figure supplement 1.** Rbfox2 expression is regulated by TGFβ signaling pathway.

DOI: https://doi.org/10.7554/eLife.45418.023

**Figure supplement 2.** Rbfox2 promoter analysis.

DOI: https://doi.org/10.7554/eLife.45418.024

## Discussion

During embryonic development, various organ formations require precise spatial and temporal regulation of gene expression. Traditionally, developmental studies were more focused on the role of transcription factors and signaling pathways. It is only in recent years that the importance of splicing factors has been demonstrated in regulating various developmental processes. The gene regulatory network required to orchestrate neural crest development is very well defined (*Gammill and Bronner-Fraser, 2003*; *Hutson and Kirby, 2003*; *Sauka-Spengler and Bronner-Fraser, 2008*; *Simões-Costa et al., 2014*). However, the cell-autonomous role of splicing factors in neural crest development is poorly investigated. In the present study, we demonstrate a critical role for splicing regulator Rbfox2 in NCCs. We show that Rbfox2 is expressed in pre-migratory and migratory NCCs, neural crest-derived palate shelves, dorsal root ganglia, and somites. Genetic deletion of *Rbfox2* using *Pax3^{Cre/+}* or *Wnt1^{Cre/+}* mouse strains affected neural crest cells and their derivatives, causing severe craniofacial defects including cleft palate. We show that cleft palate defect was due to impaired palate cell proliferation and not due to cell death or impaired NCCs migration. To examine the effect of *Rbfox2* deletion on cranial NCCs, we examined the cranial NC-derived craniofacial skeletons and found that majority of the NC-derived bones were affected in *Rbfox2* mutants.

NCCs contribute to the formation and septation of the cardiac OFT, as well as patterning and remodeling of aortic arch arteries. In contrast to a fully penetrant cleft palate and craniofacial bone defects, no cardiac defects were observed in *Rbfox2* mutants. This surprising finding led us to investigate if Rbfox2 is expressed during OFT development. We found Rbfox2 expression was not detected in the NC-derived cardiac tissues; thereby explaining the lack of OFT defects. NCCs also contribute to the peripheral tissues such as nervous systems, thymus, adrenal gland and others (*Dupin and Sommer, 2012*). With the exception of cranial nerves, all other neural crest-derived tissues analyzed develop grossly normal, suggesting that Rbfox2 may not be or only transiently expressed in these NC-derived tissues. These findings indicate that *Rbfox2* is required for the development of a narrow subset of the NCCs.

Since *Pax3^{Cre}* is also active in non-neural crest-derived tissues such as somites and limb and diaphragm muscles (*Bajard et al., 2006*; *Engleka et al., 2005*), we included these tissues in our analysis. In contrast to the controls, ectopic bone formation and fusion of vertebral bodies was observed in *Rbfox2^{Pax3-CKO}* embryos, most likely the reason for the straight vertebral column. As the metameric organization of the axial skeleton is derived from the somites, these results demonstrate that *Rbfox2* in the somites is necessary for proper development of the vertebral column. No obvious defects in the limb and diaphragm muscles were observed. Since both *Rbfox1* and *Rbfox2* are co-expressed in skeletal muscle, it is possible that *Rbfox1* is able to compensate for the loss of *Rbfox2*, thus preventing any developmental defects. Recently, *Singh et al. (2018)* generated skeletal muscle-specific Rbfox1/2 double knockout and observed a severe reduction in muscle mass, suggesting functional redundancy.

Splicing is a tightly regulated process required for increasing the transcriptome complexity using a finite set of genes, (*Baralle and Giudice, 2017*; *Revil et al., 2010*). It enhances proteomic diversity by increasing the number of distinct mRNAs transcribed from a single gene. Depending upon the type of tissues and organs, subtle changes in mRNA by splicing may alter RNA stability and/or function, protein interaction networks by either removing or inserting protein domains, subcellular localization and gene expression (*Baralle and Giudice, 2017*; *Garcia-Blanco et al., 2004*; *Revil et al., 2010*). Genetic mutations in splicing regulators have been reported in various human diseases (*Cieply and Carstens, 2015*; *Garcia-Blanco et al., 2004*). Here, we not only demonstrate that *Rbfox2* is essential for the development of tissues derived from NCCs, but also uncover over 100 Rbfox2-dependent splicing events that occur during neural crest development.

RNA sequencing analysis revealed that *Rbfox2* deletion altered splicing and expression of genes involved in the neural crest or craniofacial development. For example, *Map3k7* encodes transforming growth factor β (TGF-β)-activated kinase 1 (Tak1) required for the proliferation of palatal mesenchymal cells. Neural crest-specific deletion of *Tak1* results in the cleft palate (*Song et al., 2013*; *Yumoto et al., 2013*). Fibronectin 1 (*Fn1*), which is a component of the extracellular matrix, has various alternatively spliced variants. Wang et al. recently demonstrated that neural crest-specific deletion of *Fn1* leads to the cleft palate development (*Wang and Astrof, 2016*). Reduced expression of both *Map3k7* and *Fn1* was observed in *Rbfox2* mutant tissues. We also identified a number of

candidate genes that are novel and their role in neural crest/craniofacial development has not been established. In the present study, in addition to changes in splicing, we also identified 56 genes that are differentially expressed between control and *Rbfox2* mutant embryos. A number of genes implicated in craniofacial bone development such as *Igf1*, *Wnt5a*, *Fn1*, and *Aldh1a2* etc. were downregulated in *Rbfox2* mutants (*Halilagic et al., 2007*; *Lohnes et al., 1994*; *Yamaguchi et al., 1999*; *Zhang et al., 2002*). We found that only five (*Meg3*, *Ccnl2*, *Smarca2*, *Tpm1,* and *Fn1*) of 56 differentially expressed genes were also alternatively spliced, suggesting that Rbfox2 regulates transcriptional gene networks apart from alternative splicing. This is not surprising considering Rbfox2 has been reported to regulate gene expression patterns by different mechanisms (*Damianov et al., 2016*; *Fogel et al., 2012*; *Lee et al., 2016*; *Wei et al., 2016*). For example, Rbfox2 can affect transcript stability by directly binding to the 3'UTR of target genes (*Damianov et al., 2016*; *Lee et al., 2016*). Rbfox2 may affect gene expression by recruiting polycomb complexes to the DNA (*Wei et al., 2016*). Future work in this direction will help to determine the mechanism by which Rbfox2 regulates expression of neural crest genes. Recent studies have defined the transcriptional network required for regulating many aspects of cranial neural crest biology (*Simões-Costa et al., 2014*). However, the mechanisms by which splicing factors could be integrated into these gene regulatory networks need to be further explored.

The importance of both canonical and non-canonical TGF-β signaling pathways in craniofacial development, including secondary palate formation, has been studied extensively (*Iwata et al., 2012*; *Massagué, 2012*; *Shim et al., 2009*; *Song et al., 2013*; *Yumoto et al., 2013*). Canonical TGF-β signaling occurs via activation of receptor-regulated Smads (R-Smads) 2/3. However, non-canonical TGF-β signaling occurs via activation of the mitogen-activated protein kinase (MAPK) pathway, including TGF-β-activated kinase 1 (Tak1) (*Massagué, 2012*). TGF-β signaling is required in the NCCs to regulate cell proliferation during palatogenesis (*Iwata et al., 2012*; *Song et al., 2013*). In humans, altered TGF-β signaling pathway has been associated with both syndromic and non-syndromic cleft palates. For instance, mutations in either TGF-β receptor type I (*TGFBR1*) or type II (*TGFBR2*) are associated with Loeys-Dietz syndrome (*Loeys et al., 2005*; *Mizuguchi et al., 2004*). Patients with Loeys-Dietz syndrome have craniofacial malformations, including cleft palate, craniosynostosis and hypertelorism (*Loeys et al., 2005*; *Mizuguchi et al., 2004*). Similarly, patients with Marfan or DiGeorge syndrome develop craniofacial malformations from altered TGF-β signaling (*Brooke et al., 2008*; *Kalluri and Han, 2008*; *Lindsay, 2001*; *Wurdak et al., 2005*). In mice, neural crest-specific genetic inactivation of several TGF-β receptors in mice, including *Bmprla*, *Tgfbr1*, and *Tgfbr2* causes craniofacial deformities, including cleft palate (*Dudas et al., 2006*; *Ito et al., 2003*; *Li et al., 2013*). In the present study, we observed that *Rbfox2* deletion in NCCs affected the expression and splicing of a number of genes implicated in the TGF-β signaling pathway, leading to deregulated Rbfox2-TGF-β-Tak1 signaling axis. For example, *Map3k7* encoding Tak1 is differentially spliced and downregulated at both mRNA and protein levels in *Rbfox2* mutant cells. Tak1 expression, but not its activity, was significantly reduced in the palatal mesenchyme of *Rbfox2* mutant embryos.

Recent studies in mice demonstrated that Tak1 is required in the NCCs to activate both TGF-β-induced canonical (R-Smads) and non-canonical (p38 Mapk) pathways (*Yumoto et al., 2013*). Fn1, a positive regulator of TGF-β signaling, was downregulated at both mRNA and protein levels. Fn1 and its integrin receptor positively regulate TGF-β signaling by promoting the receptor complex formation on the cell surface (*Tian et al., 2012*). Fibronectin also regulates TGF-β by controlling the matrix assembly of latent TGF-β-binding protein-1 (*Dallas et al., 2005*). In a reciprocal manner, TGF-β1 induces Fn1 expression via a Smad independent pathway (*Hocevar et al., 1999*). Meg3 modulates the expression of TGF-β pathway genes by binding to the distal regulatory elements (*Mondal et al., 2015*). Similar to *Rbfox2* mutants, neural crest-specific deletion of *Map3k7* or *Fn1* leads to craniofacial defects including cleft palate (*Song et al., 2013*; *Wang and Astrof, 2016*; *Yumoto et al., 2013*). Chondrocyte-specific deletion of Tak1 results in severe chondrodysplasia with impaired ossification and joint abnormalities including tarsal fusion (*Shim et al., 2009*). Osteoblast-specific deletion of Tak1 results in clavicular hypoplasia and delayed fontanelle fusion (*Greenblatt et al., 2010*). Interestingly, loss of *Rbfox2* results in similar defects such as fused cervical bones, hypoplastic craniofacial bone, delayed ossification and fusion of cranial bones. Consistent with these findings, heterozygous mutations in *MAP3K7* cause cardiospondylocarpofacial syndrome, as characterized by craniofacial and cardiac defects including dysmorphic facial bones and extensive posterior cervical vertebral

synostosis (*Le Goff et al., 2016*; *Wade et al., 2016*). Altogether, the striking similarities in craniofacial and skeletal phenotypes between *Tak1* and *Rbfox2* mutants suggest that Tak1 is a downstream target of Rbfox2 and it may significantly contribute to the phenotype observed in *Rbfox2* mutant embryos. Furthermore, restoration of TGF-β signaling by Tak1 overexpression can rescue the proliferation defects seen in *Rbfox2* mutant embryos. This indicates that Tak1 regulates neural crest-derived tissues downstream of Rbfox2. It is possible that other Rbfox2 target genes identified in our RNA-Seq screen may be responsible or contribute to the craniofacial phenotype present in *Rbfox2* mutant embryos. Thus, further functional characterization and investigation of their expression and splicing patterns are clearly warranted.

Since a single splicing factor affects the expression/splicing of numerous genes and displays profound downstream effects, changes in the expression levels of splicing factors must be tightly regulated during embryonic development (*Baralle and Giudice, 2017*; *Revil et al., 2010*). We found that expression of Rbfox2 in cranial NCCs is dependent on TGF-β signaling. Furthermore, Rbfox2 is required for TGF-β-Tak1 signaling axis in embryonic neural crest development. Given that altered TGF-β signaling is well studied in multiple human congenital malformations and syndromes, our observation may be relevant in human disease studies. In summary, we have provided evidence that Rbfox2 modulates neural crest development.

# Materials and methods

**Key resources table**

| Reagent type (species) or resource | Designation | Source or reference | Identifiers | Additional information |
|---|---|---|---|---|
| Genetic reagent (*M. musculus*) | *Rbfox2flox/flox* | (*Gehman et al., 2012*) | IMSR Cat# JAX:014090, RRID:IMSR_JAX:014090 | |
| Genetic reagent (*M. musculus*) | *Pax3Cre/+* | (*Engleka et al., 2005*) | IMSR Cat# JAX:005549, RRID:IMSR_JAX:005549 | |
| Genetic reagent (*M. musculus*) | *Wnt1Cre2* | (*Lewis et al., 2013*) | IMSR Cat# JAX:022137, RRID:IMSR_JAX:022137 | |
| Genetic reagent (*M. musculus*) | *R26mTmG/+* | (*Muzumdar et al., 2007*) | IMSR Cat# JAX:007676, RRID:IMSR_JAX:007676 | |
| Antibody | anti-Tak1 (mouse monoclonal) | Santa Cruz Biotechnology | Cat# sc-166562, RRID:AB_2140220 | WB (1:300) |
| Antibody | anti-Fibronectin (mouse monoclonal) | Santa Cruz Biotechnology | Cat# sc-8422, RRID:AB_627598 | WB (1:300) |
| Antibody | anti-β-actin (mouse monoclonal) | Santa Cruz Biotechnology | Cat# sc-47778, RRID:AB_626632 | WB (1:1000) |
| Antibody | anti-pTak1 (rabbit polyclonal) | Cell Signaling Technology | Cat# 9339, RRID:AB_2140096 | WB (1:500) |
| Antibody | anti-Smad2-C (rabbit monoclonal) | Cell Signaling Technology | Cat# 5339, RRID:AB_10626777 | WB (1:500) |
| Antibody | anti-pSmad2-C (rabbit monoclonal) | Cell Signaling Technology | Cat# 3108, RRID:AB_490941 | WB (1:500) |
| Antibody | anti-p38 Mapk (rabbit polyclonal) | Cell Signaling Technology | Cat# 9212, RRID:AB_330713 | WB (1:500) |
| Antibody | anti-p-p38 Mapk (rabbit monoclonal) | Cell Signaling Technology | Cat# 4631, RRID:AB_331765 | WB (1:500) |
| Antibody | anti-Rbfox2 (mouse monoclonal) | Abcam | Cat# ab57154, RRID:AB_2285090 | WB (1:500), IF (1:25) |
| Antibody | anti-Ki67 (rabbit monoclonal) | Abcam | Cat# ab16667, RRID:AB_302459 | IHC (1:100) |
| Antibody | anti-αSMA (mouse monoclonal) | Sigma-Aldrich | Cat# A2547, RRID:AB_476701 | IHC (1:100) |

*Continued on next page*

*Continued*

| Reagent type (species) or resource | Designation | Source or reference | Identifiers | Additional information |
|---|---|---|---|---|
| Antibody | anti-Neurofilament (mouse monoclonal) | DSHB | Cat# 2H3, RRID:AB_531793 | Wholemount staining (1:50) |
| Recombinant DNA reagent | pCMV5B-HA-Smad2 plasmid | Addgene | RRID:Addgene_11734 | |
| Recombinant DNA reagent | pCMV5B-Flag-Smad3 plasmid | Addgene | RRID:Addgene_11742 | |
| Recombinant DNA reagent | pCMV5B-Smad4 plasmid | Addgene | RRID:Addgene_11743 | |
| Recombinant DNA reagent | pcDNA3-TAK1/F plasmid | Addgene | RRID:Addgene_44161 | |

## Mice

$Pax3^{Cre/+}$, $Wnt1^{Cre2}$, $Rbfox2^{flox/flox}$, and $R26^{mTmG/+}$ mice were maintained on a mixed genetic backgrounds (*Engleka et al., 2005*; *Gehman et al., 2012*; *Lewis et al., 2013*; *Muzumdar et al., 2007*). $Rbfox2$ mutant mice were generated by crossing the $Pax3^{Cre/+}$ mice with $Rbfox2^{flox/flox}$ mice (*Engleka et al., 2005*; *Gehman et al., 2012*). Resulting $Pax3^{Cre/+};Rbfox2^{flox/+}$ offspring were then back-crossed to $Rbfox2^{flox/flox}$ mice to obtain $Pax3^{Cre/+};Rbfox2^{flox/flox}$ (presented as $Rbfox2^{Pax3-CKO}$ throughout the manuscript) mice. Similarly, $Wnt1^{Cre2}$-mediated neural crest-specific $Rbfox2$ mutant mice were generated by crossing the $Wnt1^{Cre2}$ mice (Jackson Laboratory, 022137) with $Rbfox2^{flox/flox}$ mice (*Gehman et al., 2012*; *Lewis et al., 2013*). Resulting $Wnt1^{Cre2};Rbfox2^{flox/+}$ offspring were then back-crossed to $Rbfox2^{flox/flox}$ mice to obtain $Wnt1^{Cre2};Rbfox2^{flox/flox}$ (presented as $Rbfox2^{Wnt1-CKO}$ throughout the manuscript) mice. Control ($Rbfox2^{flox/+}$ or $Rbfox2^{flox/flox}$ or $Pax3^{Cre/+};Rbfox2^{flox/+}$ or $Wnt1^{Cre2};Rbfox2^{flox/+}$) and mutant ($Pax3^{Cre/+};Rbfox2^{flox/flox}$ or $Wnt1^{Cre2};Rbfox2^{flox/flox}$) embryos were harvested from timed pregnancies counting the afternoon of the plug date as E0.5. Embryos were dissected in PBS and fixed in 4% paraformaldehyde (PFA) solution in PBS. Genotyping was performed on DNA isolated from either yolk sacs or tail biopsies using following primers: 5'-ATTC TCCCACCGTCAGTACG-3' and 5'-CGTTTTCTGAGCATACCTGGA-3' for $Pax3^{Cre/+}$; 5'-CAG CGC CGC AAC TAT AAG AG-3' and 5'-CAT CGA CCG GTA ATG CAG-3' for $Wnt1^{Cre2}$ and, 5'-AACAA-GAAAGGCCTCACTTCAG-3' and 5'-GGTGTTCTCTGACTTATACATGCAC-3' for $Rbfox2^{flox/flox}$. $R26^{mTmG/+}$ embryos were genotyped based on RFP expression (*Muzumdar et al., 2007*). Littermate embryos were analyzed in all experiments unless otherwise noted. The Institutional Animal Care and Use Committee (IACUC) at SingHealth and Duke-NUS Medical School approved all the animal experiments.

## Histology and immunohistochemistry

Whole embryos and isolated tissues were dissected in PBS, fixed in 4% paraformaldehyde (PFA) overnight at 4°C, followed by PBS washes and transferred to different gradients of ethanol for processing and paraffin embedding for sectioning. H and E staining was performed for gross histological analysis using standard procedures (*Katz et al., 2012*; *Singh et al., 2010*). Immunohistochemical analysis was performed on paraffin sections of PFA-fixed embryos. Primary antibodies used for whole mount or section immunohistochemistry were: anti-Rbfox2 (Fox2/RBM9) mouse monoclonal (Abcam ab57154), anti-Ki67 rabbit monoclonal antibody (Abcam, Cat. no. ab16667), anti-α-Smooth Muscle actin mouse monoclonal antibody (Sigma, Cat. No. A2547), and anti-2H3 mouse polyclonal (Iowa Hybridoma Bank, developed by T. M. Jessell and J. Dodd). Whole-mount immunostaining for neurofilament (2H3) was carried out as described previously (*Meadows et al., 2013*; *Singh et al., 2005a*; *Singh et al., 2011*). Briefly, endogenous peroxidase activity was blocked with 5% $H_2O_2$/methanol for 2 hr at room temperature. The anti-2H3 mouse polyclonal primary antibody (Iowa Hybridoma Bank, developed by T. M. Jessell and J. Dodd) was applied overnight at 4°C at a dilution of 1:200. The goat anti-mouse IgG-HRP secondary antibody (Santa Cruz, Cat. no. sc-2005) was applied overnight at 4°C at a dilution of 1:500. Detection of HRP activity was performed using a DAB kit (Vector Laboratories, SK-4100).

## Proliferation and TUNEL assay

Cell proliferation was evaluated by Ki67 immunohistochemistry (Abcam, Cat. no. ab16667) on E12.5 and E15.5 control and knockout palate sections. DAPI (Vector Laboratories) was used to stain the nuclei. For each genotype, images of 4–6 different sections of 3–4 independent embryos were used. Apoptosis was detected using In Situ Cell Death Detection Kit, Fluorescein (Roche, Cat no. 11684795910) following the manufacturer's instructions.

## Alcian blue/Alizarin red and Von Kossa staining

Alcian Blue/Alizarin Red staining of bone and cartilage was performed as described previously (*Singh et al., 2005b*). Briefly, euthanized embryos were placed in tap water for 1–2 hr at 4°C. Embryos were placed in 65°C water for 30 s allowing easy removal of skin. Visceral organs were removed under the microscope and the embryos were placed in 100% ethanol for 2–3 days at room temperature. Cartilage staining was performed using Alcian blue solution (150 mg/L Alcian blue 8GX in 80% ethanol/20% acetic acid) for 2–3 days. Embryos were rinsed and post-fixed overnight in 100% ethanol. Bone staining was performed using Alizarin red solution (50 mg/L Alizarin red S in 0.5% KOH) for 1–2 days at room temperature. Embryos were incubated in 0.5% KOH until most of the soft tissues were digested. The 0.5% KOH solution was replaced with 20% glycerol in water and incubated at room temperature until tissues cleared completely. Imaging was done using an inverted Olympus dissecting microscope.

## RNA sequencing, data analysis and, RT-PCR validation

Craniofacial tissue was microdissected from E12.5 control and knockout embryos in cold PBS. Three independent biological replicates were used for each genotype group. Tissues were homogenized and RNA was isolated using a PureLink RNA Mini kit from Thermo Fisher (Cat. no. 12183018A). Sequencing libraries of poly(A)+RNA from 3 control and three mutant samples were prepared using the TruSeq Stranded mRNA Library Prep Kit (Illumina) according to manufacturer's instructions. Biological replicates were individually barcoded and pooled for paired-end sequencing using Illumina HiSeq4000 platform at the Genome Institute of Singapore. For each sample, approximately ~60–80 million paired-end reads of 151 bp were used for genome-guided alignment. Paired-end fastq sequence reads from each sample were aligned to mouse reference genome (GRCm38) using ultrafast RNA-seq aligner STAR (*Dobin et al., 2013*) with 82% average mapping rate and negligible ribosomal RNA contamination (<1%). Differential expression of genes and transcripts between controls and knockout samples were determined using two tools: MISO (Mixture of Isoforms) (*Katz et al., 2010*) and Cuffdiff (*Trapnell et al., 2013*). For Cuffdiff analysis, transcripts with FPKM $\geq$5 either in all control or all mutant samples were retained for further analysis. A similar filtering was employed for MISO where transcripts with assigned count $\geq$10 either in all control or all mutant samples were retained. Transcripts with log2 (fold change)>1 or $\leq$-1 and nominal p$\leq$0.05 were considered differentially expressed in Cuffdiff. Transcripts with delta PSI (percent spliced in)>0.2 or<−0.2 and Bayes factor $\geq$1 were similarly considered differentially expressed in MISO. The overlap of differentially expressed transcripts representing alternative splicing events between Cuffdiff and MISO was visualized via Venn diagrams (http://bioinfogp.cnb.csic.es/tools/venny/). For selected genes, Sashimi plots (https://software.broadinstitute.org/software/igv/Sashimi) were generated in MISO, depicting the distribution of raw RNA-Seq densities mapped to the exons and splice junctions of gene isoforms across control and mutant samples. A number of alternatively spliced genes were validated by RT-PCR as described previously (*Singh et al., 2016*). Briefly, for cDNA synthesis 1 ug of total RNA was used from craniofacial tissue samples. RNA was reverse-transcribed using random hexamer primed M-MLV reverse transcriptase (Promega, Madison, WI).

Primers used for RT-PCR analysis to detect splicing changes in Rbfox2 target genes are listed below.

Map3k7 Exon11-F: GAGCTTGGGAGCCTCTCGTG
Map3k7 Exon13-R: GGTTCTGTCCCAGTAACAGTC
Fn1 Exon21-F: GAGGTGACAGAGACCACAATTG
Fn1 Exon23-R: GTAAGCACTCCAGTGTCAGG
Uap1 Exon2-F: CGCACGAATGGAGCCTGTG
Uap1 Exon4-R: AACTCCTTCGTTGATTCCATTG

Postn Exon16-F: GTTCGTGGCAGCACCTTCAAAG
Postn Exon18-R: CCGTGGATCACTTCTGTCACCG
Primers used for RT-PCR analysis are listed below.
Map3k7 F: GTTCAAACCGAAATCGCATTG
Map3k7 R: CTTGTCGTTTCTGCTGTTGGC
Fn1 F: GAAGCAACGTGCTATGACGATG
Fn1 R: GTCTCTGTCAGCTTGCACATC
Smarca2 F: CTCCTGGACCAATTCTGGGG
Smarca2 R: CATCGTTGACAGAGGATGTGAG
Myl1 F: AAGATCGAGTTCTCTAAGGAGCA
Myl1 R: TCATGGGCAGAAACTGTTCAAA
Sfrs18 F: GGAGCAGTTCCGAATCCCC
Sfrs18 R: GCCTTCTTACCAGACCTTTGAG

## Primary palatal mesenchymal cell cultures

Palate shelves were dissected from E14.5 control and knockout embryos in cold PBS. Palate shelves were homogenized and plated on gelatin-coated culture plates. After 6–8 days, the cells were stimulated with 10 ng/ml recombinant TGFβ growth factors (PeproTech Cat no. #100–21) at different time points and harvested for western blot analysis. The neural crest origin and purity of cultures were confirmed by establishing cultures of embryos that carried a $Pax3^{Cre/+}$ knock in and a $Rosa26^{mTmG/+}$ reporter ($Pax3^{Cre/+}$; $Rbfox2^{flox/+}$; $Rosa26^{mTmG/+}$). Majority of cultured cells are GFP positive demonstrating their neural crest origin. For inhibitor experiments, primary palatal mesenchymal cells were seeded with a density of $2 \times 10^6$ cells/ml onto a 6-well plate and cultured in DMEM supplemented with 1% penicillin/streptomycin and 10% FBS. After 48 hr, cells were washed with PBS and starved overnight in basal medium supplemented with 1% FBS. Cells were then stimulated with/without recombinant TGFβ (10 ng/ml) in the presence/absence of SB203580 (5, 10 and 20 µM) (Selleckchem Cat no. #S1076), SIS3 (1, 5 and 10 µM) (Selleckchem Cat no. #S7959) or 5Z-7-Oxozeaenol (0.1, 1.0 and 5 µM) (R and D Systems Cat no. #3604) for desired time period (6 hr and 24 hr) and harvested for western blot analysis. For Tak1 rescue experiment, primary palatal mesenchymal cells were isolated from E14.5 control and knockout embryos and upon reached ~80% of confluence, knockout cells were transfected with control vector (pcDNA3) or pcDNA3-TAK1/FLAG (Addgene, Plasmid #44161) using Lipofectamine 2000 reagent (Thermo Scientific, catalog no. 11668–027), according to manufacturer's protocol. Seventy-two hours after the transfection, cells were fixed with 4% paraformaldehyde and processed for Ki67 immunostaining. In a separate experiment, cell lysate was collected to detect the TAK1 and pTAK1 by western blot analysis.

## Western blot analyses

Micro-dissected palate shelves or cultured palate mesenchymal cells were washed with DPBS and lysed with RIPA buffer (Thermo Scientific, catalog no. 89901) containing 1:100 diluted protease and phosphatase inhibitor cocktail (Sigma). The cell lysates were centrifuged at 13,000 rpm for 10 min at 4°C and the supernatants were collected for immunoblot analyses. Total protein concentration was determined by using the Pierce BCA protein assay kit (Thermo Scientific, catalog no. 23225). Western blots were performed as described previously (Singh et al., 2019; Singh et al., 2016). Briefly, for western blotting, 20–25 µg of total protein samples were separated by SDS-PAGE and transferred to nitrocellulose membrane using a Trans-Blot Turbo system (Bio-Rad). Membranes were then blocked with 2–5% BSA in TBS containing 0.1% Tween (TBST) and subsequently incubated with primary antibodies diluted in TBST containing 2–5% BSA for overnight at 4°C. Blots were then washed in TBST and incubated for 1.5 hr at room temperature with the appropriate horseradish peroxidase-linked secondary antibodies (Santa Cruz). Immunoreactive bands were detected by chemiluminescence (Hiss GmbH, catalog no. 16026) using Gel Doc XR + System (Bio-Rad). Primary antibodies used were as follows: anti-Tak1 (1:300; Santa Cruz sc-166562), anti-pTak1 (1:500; Cell signaling 9339S), anti-Smad2-C (1:500; Cell signaling 5339), anti-pSmad2-C (1:500; Cell signaling 3108), anti-p38 Mapk (1:500; Cell signaling 9212), anti-p-p38 Mapk (1:500; Cell signaling 4631), anti-Fibronectin (1:300; Santa Cruz sc-8422), anti-Rbfox2 (1:500; Abcam ab57154) and anti-β-actin (1:1000; Santa Cruz sc-47778).

## Plasmids

Mouse Rbfox2 promoter (~1.7 kb) was amplified and cloned into pGL4.27 vector (Promega) using In-Fusion HD Cloning Kit (Clontech Cat no. 639645) for the luciferase assays. Expression vectors pCMV5B-HA-Smad2 (Addgene plasmid # 11734) (*Eppert et al., 1996*), pCMV5B-Flag-Smad3 (Addgene plasmid # 11742)(*Labbé et al., 1998*) and pCMV5B-Smad4 (Addgene plasmid # 11743)(*Macías-Silva et al., 1996*) were a gift from Jeff Wrana. Expression vector pcDNA3-TAK1/F was a gift from Xin Lin (Addgene plasmid # 44161) (*Blonska et al., 2005*).

## Luciferase assay

Luciferase assay was performed as previously described (*Singh et al., 2016*; *Singh and Epstein, 2012*). HEK293T cells are the most commonly used cell line for monitoring the activity of the TGF/SMAD signaling pathway. Briefly, HEK293T cells were seeded in 12-well plates for 24 hr before transfection. The Rbfox2 luciferase reporter plasmid along with other indicated plasmids (Smad2, Smad3 or Smad4) was co-transfected using FuGENE6 reagent (Promega, catalog no. E2691). To normalize transfection, 50 ng of lacZ expression plasmid was also transfected together with other indicated plasmids. Cell extracts were prepared 60 hr post-transfection using lysis buffer (Promega, catalog No. E3971). Luciferase activities were assayed using Luciferase Reporter Assay System kit (Promega, catalog no. E1500). Lysates were also assayed for b-galactosidase activity using the b-Galactosidase Enzyme Assay System (Promega, Cat. no. E2000). Luciferase reporter activity was normalized to b-galactosidase activity. The luciferase assay results were reproduced in at least three independent experiments. All experiments were performed in duplicate, and the representative data are shown in the bar graphs.

## ChIP and RNA immunoprecipitation assays

ChIP experiments were performed as previously described (*Singh et al., 2016*). ChIP assay was performed on either unstimulated or TGFb-stimulated palate mesenchymal cells using Smad2/3 antibody (Abcam, Cat. no. ab207447), according to Millipore Chip Assay Kit protocol with minor modifications (Catalog no #17–295). RNA-IP experiments were performed as previously described with minor modifications (*Niranjanakumari et al., 2002*).

## Statistical analysis

Statistical analyses were performed using the two-tailed Student's t-test. Data are expressed as mean ± SD. Differences were considered significant when the p-value was <0.05. One-way analysis of variance (ANOVA) was used to assess statistical differences between groups. Significant ANOVA results were further analyzed by Bonferroni's multiple comparisons test (*, p<0.05; **, p<0.01; ***, p<0.001; *NS*, not significant).

# Acknowledgements

We would like to thank Sandip Chorghade for technical assistance. We are thankful to MKS lab members for helpful discussion. This work was supported by funds from Duke-NUS Medical School Singapore and the Goh foundation and a Singapore National Research Foundation (NRF) fellowship (NRF-NRFF2016-01) to MKS.

# Additional information

## Funding

| Funder | Grant reference number | Author |
|---|---|---|
| National Research Foundation Singapore | NRF-NRFF2016-01 | Manvendra K Singh<br>Masum M Mia<br>Shamini Guna Shekeran |
| Goh Foundation | | Dasan Mary Cibi |
| Duke-NUS Medical School Singapore | | Reddemma Sandireddy |

The funders had no role in study design, data collection and interpretation, or the decision to submit the work for publication.

## Author contributions
Dasan Mary Cibi, Masum M Mia, Shamini Guna Shekeran, Data curation, Formal analysis, Validation, Investigation, Methodology; Lim Sze Yun, Data curation, Validation, Investigation, Methodology; Reddemma Sandireddy, Data curation, Formal analysis, Validation, Methodology; Priyanka Gupta, Data curation, Formal analysis, Validation, Investigation; Monalisa Hota, Data curation, Formal analysis, Methodology; Lei Sun, Resources, Methodology; Sujoy Ghosh, Data curation, Formal analysis, Investigation, Methodology; Manvendra K Singh, Conceptualization, Resources, Data curation, Formal analysis, Supervision, Funding acquisition, Investigation, Visualization, Methodology, Writing—original draft, Project administration, Writing—review and editing

## Author ORCIDs
Lei Sun (iD) http://orcid.org/0000-0003-3937-941X
Manvendra K Singh (iD) https://orcid.org/0000-0002-2884-0074

## Ethics
Animal experimentation: The Institutional Animal Care and Use Committee (IACUC) at SingHealth and Duke-NUS Medical School approved all the animal experiments (IACUC protocol number 2014/SHS/0988 and 2018/SHS/1415).

## Decision letter and Author response
Decision letter https://doi.org/10.7554/eLife.45418.029
Author response https://doi.org/10.7554/eLife.45418.030

# Additional files

## Supplementary files
• Transparent reporting form
DOI: https://doi.org/10.7554/eLife.45418.025

## Data availability
RNA sequencing data have been deposited in GEO under accession code GSE127245.

The following dataset was generated:

| Author(s) | Year | Dataset title | Dataset URL | Database and Identifier |
| --- | --- | --- | --- | --- |
| Cibi DM, Mia MM, Shekeran SG, Yun LS, Sandireddy R, Gupta P, Hota M, Seshachalam VP, Sun L, Ghosh S, Singh MK | 2019 | Neural crest-specific deletion of splicing factor Rbfox2 leads to craniofacial abnormalities including cleft palate | https://www.ncbi.nlm.nih.gov/geo/query/acc.cgi?acc=GSE127245 | NCBI Gene Expression Omnibus, GSE127245 |

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
