## [Decision Letter]

Thank you for submitting your article "Neural crest-specific deletion of splicing factor Rbfox2 leads to craniofacial abnormalities including cleft palate" for consideration by *eLife*. Your article has been reviewed by two peer reviewers, and the evaluation has been overseen by a Reviewing Editor and K VijayRaghavan as the Senior Editor. The following individuals involved in review of your submission have agreed to reveal their identity: Hiroki Kurihara (Reviewer #1).

The reviewers have discussed the reviews with one another and the Reviewing Editor has drafted this decision to help you prepare a revised submission.

This article focuses on the splicing factor Rbfox2 and its expression in neural crest cells. Lineage specific gene ablation results in craniofacial bone defects and cleft palate. RNA Seq analysis identifies transcripts that are affected and in particular highlights the role of TGF-β-*Tak1* signalling in the mutant phenotype. The Rbfox2-TGF-β-*Tak1* signalling axis undergoes positive feedback regulation by TGF-β signalling on Rbxfox2.

These are interesting findings. However the following points should be addressed before publication.

1) Transcript profiles of mutant cells before and after *Tak1* rescue should be shown. Ideally this should be based on RNASeq, but if these data are not available at least quantitation by qRT-PCR analysis of transcripts shown in Figure 5 should be included.

2) The Western blot data require more rigorous quantification based on multiple independent samples, presumably underlying the representative images shown.

3) Results on the cleft palate phenotype after *Tak1* rescue should be shown, or if not observed should be commented in the Discussion.

4) In Figure 3L, M, the authors show E-cadherin expression in control and mutant mice, and state that EMT was not affected due to loss of Rbfox2. However, EMT in palate formation occurs in medial edge epithelial cells after palatal fusion in the midline. Without this midline fusion in Rbfox2 mutants, no definite conclusion can be drawn for EMT.

5) The question of whether hypoglossal nerve defects are secondary to defects in the hypoglossal cord (where Pax3 is expressed) should be discussed.

Concerning other cranial nerves, profound ophthalmic and trochlear nerves appear to be deformed, as observed in the supplemental figures?

6) Rbfox2 acts as a transcriptional regulator as well as its role in controlling splicing. Can the authors distinguish these effects? Some comment is required.

---

## [Author Response]

This article focuses on the splicing factor Rbfox2 and its expression in neural crest cells. Lineage specific gene ablation results in craniofacial bone defects and cleft palate. RNA Seq analysis identifies transcripts that are affected and in particular highlights the role of TGF-β-Tak1 signalling in the mutant phenotype. The Rbfox2-TGF-β -Tak1 signalling axis undergoes positive feedback regulation by TGF-β signalling on Rbxfox2.These are interesting findings. However the following points should be addressed before publication.1) Transcript profiles of mutant cells before and after Tak1 rescue should be shown. Ideally this should be based on RNASeq, but if these data are not available at least quantitation by qRT-PCR analysis of transcripts shown in Figure 5 should be included.

We would like to thank the reviewer for his/her comments. We have performed the *Tak1* rescue experiment again and isolated RNA for qRT-PCR analysis. To determine the effect of *Tak1* overexpression on the expression of Rbfox2-dependent genes, we performed qRT-PCR for Map3k7, Fn1, Myl1, Sfrs18 and Smarca2 on control and Rbfox2 mutant cells transfected with empty plasmid vector or a plasmid expressing *Tak1*. We observed a significant increase in the expression of Map3k7 and Fn1 in *Tak1* overexpressing Rbfox2 mutant cells compared to the empty vector transfected controls. However, no change in the expression of Myl1, Sfrs18, and Smarca2 was observed. These findings have been included in Figure 6—figure supplement 2 and discussed in the Results section.

2) The Western blot data require more rigorous quantification based on multiple independent samples, presumably underlying the representative images shown.

We have quantified all the western blots used in this manuscript. Quantification results have been included in the revised Figure 6 and 7.

3) Results on the cleft palate phenotype after Tak1 rescue should be shown, or if not observed should be commented in the Discussion.

We would like to clarify that our experiment was designed to rescue the proliferation defect observed in palate cells due to Rbfox2 deletion. We cultured the control and Rbfox2 mutant palate cells and transfected the Rbfox2 mutant cells with either empty plasmid (pcDNA3) or a plasmid expressing *Tak1* (pcDNA3-*Tak1*), and performed Ki67 immunostaining. We observed increased expression of both *Tak1* and phosphorylated *Tak1* after *Tak1* transfection in Rbfox2 mutant cells. Our results demonstrate that *Tak1* overexpression can rescue the proliferation defects observed in Rbfox2 mutant palate cells. We did not perform palatal shelf explant culture experiments to rescue palate growth and fusion.

4) In Figure 3L, M, the authors show E-cadherin expression in control and mutant mice, and state that EMT was not affected due to loss of Rbfox2. However, EMT in palate formation occurs in medial edge epithelial cells after palatal fusion in the midline. Without this midline fusion in Rbfox2 mutants, no definite conclusion can be drawn for EMT.

We agree with the reviewer. We have modified the text to reflect this.

5) The question of whether hypoglossal nerve defects are secondary to defects in the hypoglossal cord (where Pax3 is expressed) should be discussed.Concerning other cranial nerves, profound ophthalmic and trochlear nerves appear to be deformed, as observed in the supplemental figures?

We agree with the reviewer that it is possible that hypoglossal nerve defects are secondary to defects in the hypoglossal cord. Cells from occipital somites (myotomes) grow towards the tongue as the "hypoglossal cord", which arrives prior to the hypoglossal nerve. Any defects in the hypoglossal cord could affect the formation of the hypoglossal nerve. Bajard et al., have demonstrated that hypoglossal cord is derived from Pax3 expressing cells (Bajard et al., 2006). We have discussed this point in the revised manuscript. Yes, we also see changes in the structure of ophthalmic and trochlear nerves in Rbfox2 mutants. Thanks for bringing this to our attention. We have revised the Figure 4—figure supplement 3 and included these results in the revised manuscript.

6) Rbfox2 acts as a transcriptional regulator as well as its role in controlling splicing. Can the authors distinguish these effects? Some comment is required.

Differential expression of genes and transcript isoforms between controls and Rbfox2 mutant samples were determined using two tools: MISO (Mixture of Isoforms), which quantitates the expression level of alternatively spliced genes and identifies differentially regulated isoforms or exons across samples and Cuffdiff, which finds significant changes in transcript expression, splicing, and promoter use. Using these tools, we have identified over 100 alternatively spliced transcripts (Figure 5B-C) and 56 genes that are differentially expressed (Figure 5K) between control and Rbfox2 mutant embryos. We found that only 5 out of 56 differentially expressed genes were also alternatively spliced suggesting that Rbfox2 regulates transcriptional gene networks apart from alternative splicing. These details have been included in the manuscript.